

# Post-hercynian ultra-high temperature tectono-metamorphic evolution of the Middle Atlas lower crust (Central Morocco) revealed by metapelitic granulites xenoliths

Abdelkader El Maz[1], Alain Vauchez[2], Jean Marie Dautria[2]

[1]Department of Geology, University Moulay Ismaïl, Meknes, Morocco
[2]Geosciences Montpellier, Montpellier University and CNRS, Montpellier, 34095, France

*Correspondence to*: Abdelkader El Maz (elmazabdel@yahoo.fr)

**Abstract.** The study of metapelitic sillimanite- and garnet-bearing granulite xenoliths brought to the surface by the basanite of the 650 ka Tafraoute maar shed new light on the lower crust of the Tabular Middle Atlas (Morocco). Two main types of
granulites are distinguished: (1) layered quartzo-feldspathic and (2) unlayered restitic. Mineralogy, petrology, P-T estimates and EBSD data support that these granulites underwent two successive tectono-metamorphic events, before their entrapment in lava. During the first event, probably the Hercynian orogeny, the Tafraoute lower crust acquired its foliation and primary paragenesis likely including kyanite: it yields P, T conditions of 1.1 ± 0.1 GPa and 850-880 °C. The second event corresponds to a reheating up to ultrahigh temperatures (1050 ± 50 °C) under slightly lower pressure conditions (0.9 ± 0.1 GPa). This led
first to the transformation of kyanite into large prismatic sillimanite. The latter displays uncommon evidence of dislocation-creep deformation of moderate intensity that points to a tectonic episode occurring after their formation. After deformation has stopped, a reaction between sillimanite and garnet resulted in the crystallization of orthopyroxene and spinel deformation-free coronas around garnets. Approaching the peak of temperature, anhydrous partial melting of quartzo-feldspathic layers likely occurred and the resulting felsic melt spread into the rocks. This reheating event might be the consequence of the Late Permian
to Mid-Jurassic rifting that preceded the formation of the Middle Atlas range, possibly associated with underplating of hot gabbroic magma. This event was followed by gradual cooling down to ~800 °C, leading to static crystallization of the felsic melt in the quartzo-feldspathic granulites. The last event susceptible to have affected the lower crust is the alkali magmatism active in the Middle Atlas during the Mio-Plio-Quaternary. In this context, the origin of restitic granulites is questionable. It may result either from the thermal event associated to the pre-alpine rifting, or from the emplacement of basaltic dykes in the
lower crust before the quaternary eruption of the Tafraoute volcano. During this eruption, the studied granulites were entrapped in the ascending lava and very quickly transferred up to the surface, triggering the formation of small vesicular glass pockets. This study highlights the contrasted post-Hercynian evolution of the lower crust in the northern coastal alpine orogen (Rif) and the Tabular Middle Atlas: the first one underwent a tectonic exhumation without reheating during the Alpine orogeny, while the second one is characterized by a reheating to ultra-high temperature, probably during the pre-alpine rifting, but was probably
not or very little affected by the alpine events.





**Keywords:** Lower crust, metapelitic granulite xenoliths, ultra-high-temperature metamorphism, sillimanite ductile deformation, Middle Atlas, Morocco

## 1 Introduction

The current knowledge of the lower crust of the Moroccan Mediterranean margin is based on a single occurrence of kyanite-bearing granulites outcropping in the Beni Bousera massif. This massif, located about 100 km SE of Tangier (Fig.1), belongs to the Rif Coastal Alpine Range (North Morocco). The only window on the lower crust between the Mediterranean coast and the Anti-Atlas Range (~400 km south of the Rif) corresponds to granulite xenoliths sampled by the Plio-Quaternary alkali-basalts of the Middle Atlas (~200 km south of the Rif, Fig. 1). One tuff ring of this magmatic district (Tafraoute Maar, Fig.1)

is particularly rich in prismatic sillimanite-bearing granulite xenoliths. We have performed a new study of these granulites for two reasons:  1) their Ultra-High Temperature (UHT) parageneses, and 2) their major importance for the evolution of the deep crust of North Africa. Moukadiri and Bouloton (1998) have reported that these granulites might have been equilibrated at temperature >900 °C and, thus, that the lower crust likely underwent UHT conditions. However, these metamorphic conditions are still poorly constrained and their relationships with deformation and geodynamic evolution of North Africa remain

unknown.

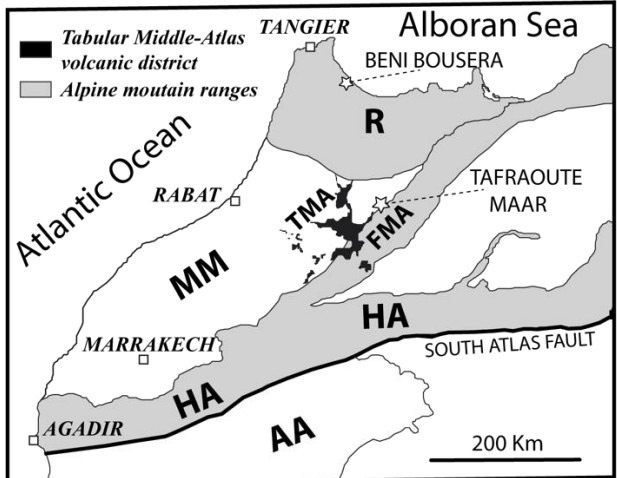

**Figure 1: Sketch map showing the location of the Tafraoute maar and the Beni Bousera massif with regard to the main mountain ranges of Morocco (AA and MM: Hercynian Anti-Atlas and Moroccan Meseta; HA, FMA, TMA and R: alpine ranges of the High Atlas, Folded Middle Atlas, Tabular Middle Atlas and Rif respectively.**


Newton and Perkins (1982) and Bohlen (1991) have suggested that the estimated peak metamorphic conditions for granulitic formations cluster within a relatively narrow field of 0.65 to 0.9 GPa and 700 to 850 °C, near the kyanite-sillimanite



equilibrium boundary. Based on a large set of granulites from various locations around the world, Harley (1989) emphasized that more than 50 % of the granulites fall outside the P-T field defined above, especially with temperatures higher than 900 °C.

For instance, ultrahigh-temperature granulites (>900 °C; 0.7-1.3 GPa) have been described in more than 50 localities around the world (e.g., Harley, 1998; Ouzegane et al. 2003a-b; Baldwin et al., 2005; Kelsey and Hand, 2015). UHT metamorphism is principally characterized by: a) high $Al_2O_3$ contents (8-12 wt %, Harley 1998) in orthopyroxene (Opx) coexisting with garnet (Grt) and sillimanite (Sil) or sapphirine, and 2) the presence of mesoperthitic feldspar and high Zr contents in rutile (Rt). Diagnostic mineral assemblages such as sapphirine + quartz (Qz), Opx + Sil + Qz, spinel (Sp) + Qz, corundum (Crn) + Qz,

and the presence of osumilite (Os) also support UHT conditions (e.g., Hensen and Green, 1973; Guiraud et al. 1996a-b; Harley, 1998, 2008; Kelsey and Hand, 2015). A pending question is how the lowermost continental crust may reach such high temperatures for such moderate pressures considering that the temperature at the base of the crust is buffered around 850 °C by fluid-absent melting. In refractory granulites, UHT metamorphism can easily develop if there is a significant heat supply, for instance due to emplacement of a large volume of mafic magma (Raith et al., 1997; Martignole and Martelat, 2003; Barbosa

et al, 2006).

The study presented here combines petrography, mineral chemistry, thermo-barometric estimates and crystallographic preferred orientation (CPO) measurements. The results of this study shed new light on the nature and evolution of the lower crust of this part of the Middle Atlas Alpine Range since the Hercynian times.

## 2   Geological setting

In Morocco, Hercynian metamorphic rocks are exposed in the North (Rif) and in the Center (Meseta). In the Meseta, the highest metamorphic ones consist of Grt + staurolite  + kyanite (Ky)-bearing pelitic micaschists, corresponding to a medium-grade Barrovian-type metamorphism (Piqué and Michard, 1989; Hoepffner et al., 2005; Chopin et al. 2014). In the Rif, which is an alpine orogenic belt resulting from the Alboran Terrane-Africa miocene collision, slices (about 150 m thick) of lower crustal granulites were exhumed simultaneously with the Beni Bousera peridotites massif (Kornprobst, 1969; El Maz and Guiraud,

2001; Álvarez-Valero et al. 2014). The primary paragenesis of Beni Bousera granulites (Grt + Ky + Rt + biotite (Bt) + plagioclase (Plg) + K-feldspar (KF) + Qz + monazite + zircon) was dated at 284 ± 27 Ma (U-Th-Pb in monazite included in Grt; Montel et al., 2000), 286-264 Ma and 300-290 Ma (zircon U-Pb dating; Melchiorre et al., 2017; Rossetti et al. 2020). The P, T equilibrium conditions of this paragenesis were estimated at 1.2 GPa and 850 °C (El Maz and Guiraud, 2001, Álvarez-Valero et al., 2014; Rossetti et al., 2020).

Two hundred km south of the Rif, the Middle Atlas range (MA, Fig. 1) is almost entirely made of Liassic to mid-Jurassic dolomitic limestones lying on a thick Triassic clay-evaporite series locally associated with doleritic basaltic flows as in central Morocco (Fiechtner et al., 1992). The MA, like the High Atlas Range further south, was formed during the alpine orogeny through the inversion of a pre-existing Triassic to Jurassic rifts system linked to the Central Atlantic Ocean opening (e.g., Mattauer et al., 1977; Frizon de Lamotte et al., 2000; Michard et al. 2008, and references therein). A NE-SW trending major



transcurrent fault (North Middle Atlas Fault) separates the MA in two domains: a folded one and a tabular one, respectively located SE and NW of the fault (Fig. 1). During the Cenozoic, the MA underwent an uplift probably related to a mantle upwelling and a lithospheric thinning (e.g., El Messbahi et al., 2015 and references therein). Mio-Plio-Quaternary alkali basaltic activity coeval with this upwelling occurred in this region (El Azzouzi et al., 2010). The most recent volcanic edifices (23 strombolian cones and 30 hydrovolcanic edifices, El Messbahi et al., 2020) are clustered within an area of ~500 km$^2$ located in

the core of the tabular domain (Fig. 1). Many of them contain xenoliths of lithospheric mantle (e.g., El Messbahi et al., 2015 and references therein) and some contain xenoliths of lower crust (Moukadiri and Bouloton, 1998). The Tafraoute maar (33°31'29''-4°41'35W; 550 ka, El Messbahi et al., 2020), where the granulitic xenoliths of metapelitic origin studied here have been collected, is located northeastward of the main volcanic area (Fig. 1). This volcanic edifice corresponds to a semi-circular tuff-ring about 2 km long and 500 m wide, deposited around a shallow explosion crater (El Messbahi et al., 2020). The

Tafraoute tuffs contain a large variety of xenoliths in addition to metapelitic granulites: (i) partially recrystallized garnet-rich metadiorites free of aluminosilicate, some of them containing minor biotite, (ii) undeformed to weakly deformed alkali to transitional gabbros similar to those found in the Central High Atlas and dated to the Mesozoic (Haillwood and Mitchell, 1971; Smith and Pozzobon, 1979) and (iii) ultramafic rocks (kaersutitite, websterite, diopsidite and spinel peridotite (El Messbahi et al., 2015). Euhedral megacrysts kaersutite (up to 10 cm long), a phase typically associated with basanitic magmatism, are

abundant. In addition, dark blue corundum euhedral crystals (up to 2 cm large) are frequent. Such crystals are usually considered as resulting from the mixing of felsic melt with a Si-poor magma (Guo et al., 1996; Sutherland et al., 1998; Giuliani et al., 2009).

In this work, the petrological characteristics of the granulites from the Tafraoute maar will be compared with those from the Beni Bousera massif. Determining similarities and differences of the granulites from these two sites would allow to compare

the tectono-metamorphic evolution of the lower crust of the Tabular MA with the better-known Rif one.

## 3 Sampling and analytical methods

Several tens of metapelitic granulite xenoliths, between 10 and 75 cm in size, have been collected in the Tafraoute tuffs (Fig. 2). They can be macroscopically divided in two groups on the basis of mineralogy and structure: quartzo-feldspathic granulites (QFG) with a well-marked layering (Fig. 2a) and restitic ones (RG) almost unlayered and free of quartz and feldspars (Fig. 2b).

Thin sections have been prepared for a selection of samples and studied under the optical microscope. From this study, five metapelitic xenoliths representative of the lower crust of Tafraoute have been selected for Electron-Back Scattering (EBSD) and Microprobe analysis: three QFG (G2, 50 cm in size; G3, 75 cm; G4, 10 cm) and two RG (TAF500 and TAF501, ~10 cm).



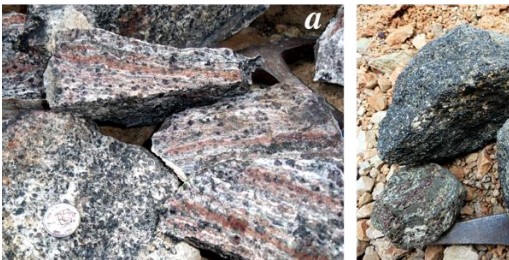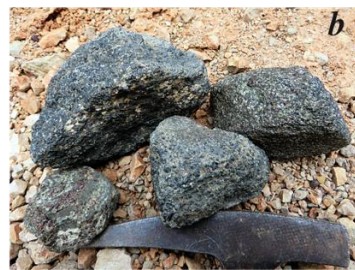

**Figure 2: Pictures of quartzo-feldspatic (a) and restitic (b) granulite samples collected in the Tafraoute tuffs. The coin diameter on (a) is 2 cm.**

Major elements composition of rock-forming minerals was analyzed at 'Microsonde Sud' facility (University of Montpellier, France) using a CAMECA-SX100 electron microprobe equipped with five wavelength-dispersive spectrometers. Operating conditions comprised an acceleration voltage of 20 kV and a 10 nA beam current. The counting time was 15 minutes for Zr in rutile and Ti in zircon, and 3 minutes for major elements.

Crystallographic mapping and preferred orientation measurements were performed on the five selected samples using indexing of EBSD patterns obtained in a CamScan Crystal Probe X500FE Scanning Electron Microscope (Geosciences Montpellier, University of Montpellier, France) equipped with an EBSD and an EDS detectors, both controlled by the AZtecHKL acquisition software (Oxford Instruments). Diffraction patterns acquisition was produced with an acceleration voltage of 17 to 21kV and a working distance of ~25mm. For each sample, EBSD data covering most of the thin section were acquired with a regular grid step between 12 and 16 µm depending on the grain size. In addition, a few more detailed maps of smaller areas have been acquired (step size between 3.5 and 5.5 µm) to better characterize the relationships between phases in destabilization coronas around garnets. Raw indexation rate varies from 74 to 80 %. Post-acquisition data processing was performed following the procedure described in detail in Baptiste et al. (2015) using the Channel 5 software (Oxford Instruments) and the MTEX toolbox in MATLAB (http://mtex-toolbox.github.io/; Hielscher and Schaeben, 2008; Bachmann et al., 2010; Bachmann et al., 2011; Mainprice et al., 2014), allowing improving the indexation rate. EBSD maps were systematically checked against microscope observations to avoid over-extrapolation. Pole figures have been generated and rotated using MTEX. The orientation distribution functions (ODF) were calculated using the "De La Vallée Poussin" kernel function with a half-width of 10°. Pole figures have been plot using the mean orientation for each grain instead of the whole pixels measurements to avoid over-representation of large crystals in the CPO.

## 4 Structure, modal composition and mineral chemistry

EBSD maps have been performed for the 5 selected granulites (Fig. 3). All studied samples contain prismatic sillimanite, garnet, orthopyroxene, spinel, rutile, ilmenite, graphite and zircon. Grt is variably destabilized and surrounded by a kelyphitic



corona. Quartz and feldspars are abundant in the QFG, but scarce or lacking in the RG. In addition, osumilite and corundum

are present but rare: the first one was observed in some samples both QFG and RG, while the second one was only found in

some RG. Brown glass is abundant in the RG samples and only in minor amount in the QFG.

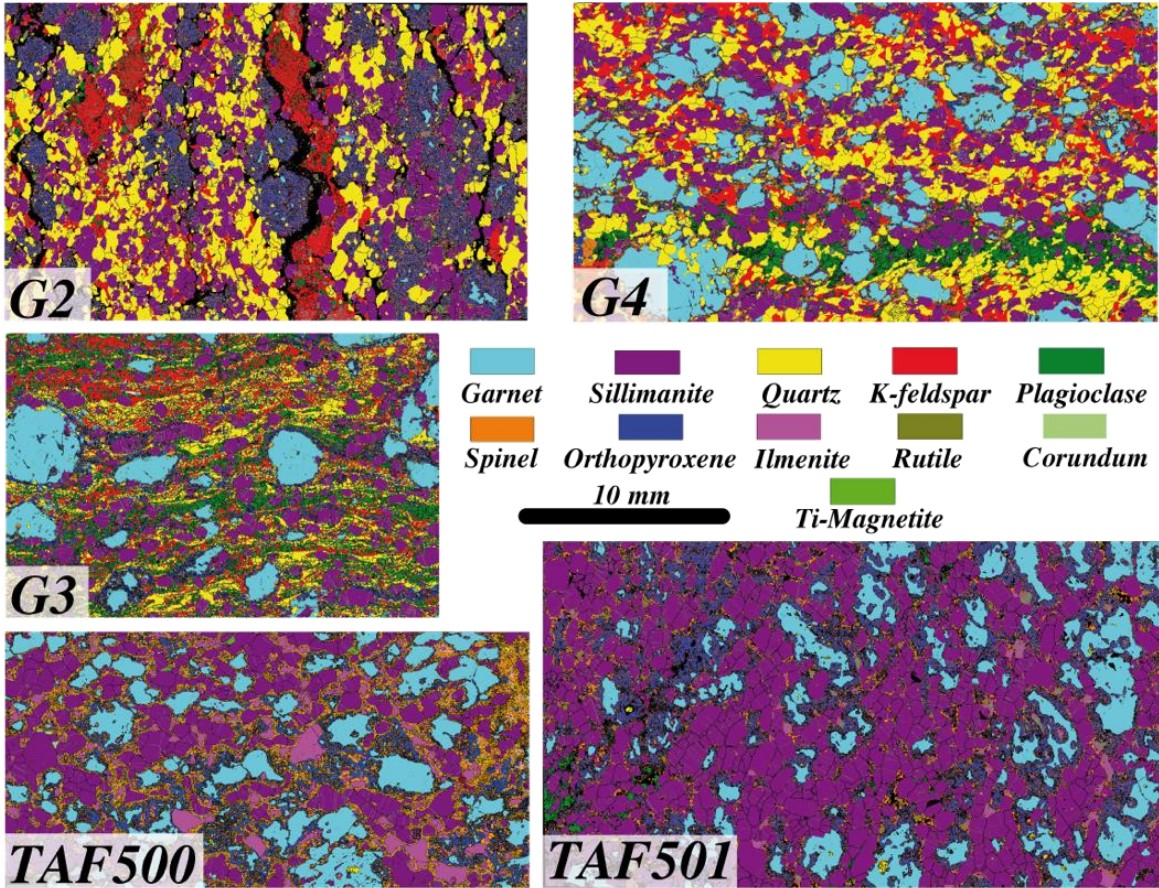

**Figure 3: EBSD maps of the five selected granulite samples. All maps are at the same scale. The variation of the maps size only depends on the mapped area. Samples G2 and TAF500 have been cut perpendicular to the foliation and also to the lineation (YZ plane), while G3, G4 and TAF501 have been cut perpendicular to the foliation and parallel to the lineation (XZ plane). Osumilite, although present in low proportion in the quartzo-feldspathic samples, is not represented in the maps because the crystals are too small in size. Individual maps are available in supplementary**

**material.**



Modal proportions of minerals (Table 1) have been evaluated from EBSD measurements. All studied samples display a foliation marked by the shape-preferred orientation of large sillimanite crystals. In the QFG, this foliation is also underlined by sometimes almost monomineralic layers of Qz, Plg and KF.

**Table 1: Modal proportions of rock-forming minerals for all selected granulites. The values have been evaluated from EBSD maps. The "Not Indexed" component includes glass, fractures and crystals too small to be correctly indexed, such as some orthopyroxene, spinel and quartz in garnet kelyphitic coronas. Tiny osumilite grains were detected in quartzo-feldspathic granulites, but in too small proportions to be included in the mode. Small grains of corundum have been found only in the restitic granulite TAF500.**

| Samples | G2 | G3 | G4 | TAF500 | TAF501 |
|---|---|---|---|---|---|
| **Grt** *(Vol. %)* | 0.5 | 13 | 18.8 | 18.6 | 14 |
| **Sil** *(Vol. %)* | 25 | 18 | 23.4 | 28.2 | 46 |
| **Qz ; Pl ; Kfs** *(Vol. %)* | 25.4 ; 2.8 ; 11.3 | 15 ; 17 ; 9.8 | 22.1 ; 7.2 ; 8.7 | 3.3 ; <0.1 ; 0 | 0.8 ; 0.8 ; <0.1 |
| **Opx; Sp; Rt; Il; TM** *(Vol. %)* | 9.8 ; 9.6 ; <0.1 ; 1.5; 0.5 | 6 ; 5.9 ; 0.6 ; 0.4; 0 | 4.6 ; 8.4; 0.2 ; 2.1; 0.4 | 10.2 ; 19 ; <0.1 ; 8.6; 0 | 13.3 ; 8.6 ; 1 ; 1.9; 0.4 |
| **NI = Glass + ITO + Error** *(Vol. %)* | 14 | 18.1 | 4.5 | 12 | 14 |


**4.1 The layered quartzo-feldspathic granulites (QFG)**

The QFG display a large variety of modal compositions (Table 1) and microstructures (Fig. 3) essentially due to the degree of Grt breakdown and the habitus of feldspars and Qz. On the opposite, Sil displays similar characteristic in all samples. In addition, the major constitutive minerals display minor chemical variations (Tables S1 to S10).

Sil (18-25 vol. % depending on the sample) appears as large prismatic crystals (≥ 1 mm wide and up to 3 mm long) displaying evidence of dislocation-creep intracrystalline deformation (Fig. 4a): shape-preferred orientation, undulose extinction and spaced subgrain boundaries perpendicular to cleavages and to the long axes of crystals (Fig. 4a). Some crystals are bent and exhibit fan-like subgrain boundaries (Fig. 4b), and clusters of minute sillimanite new grains (~10-20 μm) occur within or at the rim of some large crystals (Fig. 4c). In several samples (e.g., G2, Fig. 3), sillimanite crystals often display serrated
boundaries underlined by a thin, discontinuous strip of interpenetrating tiny anhedral crystals of spinel (≤40 μm) and quartz (≤10 μm). Some prismatic sillimanites display two cleavages at an angle of ~74° similar to kyanite, suggesting that they represent a polymorphic species intermediate between these two aluminosilicates, as described by Leyreloup (1974), Lal et al. (1984), Raith et al. (1997) and Tong and Wilson (2006). The chemical composition of sillimanite (Table S1) is almost identical in all analyzed samples; the only noticeable peculiarity is their high Fe content (0.9 < % FeO < 1.4).




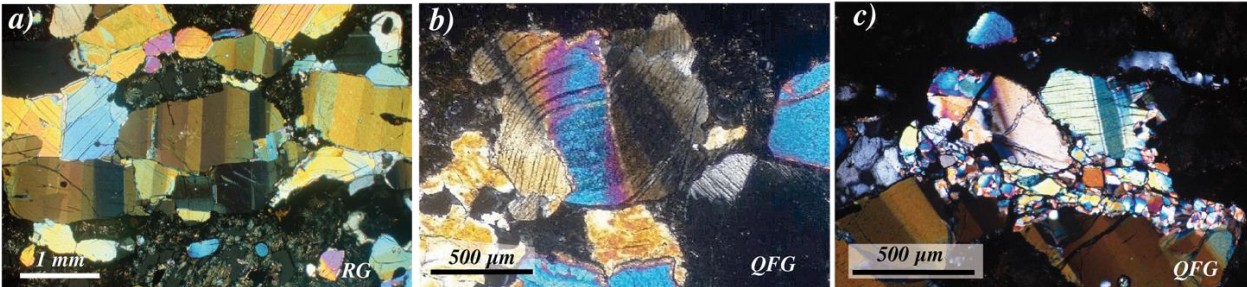

**Figure 4: Photomicrograph of deformed sillimanite grains in polarized-analyzed light. (a) In TAF501, crystals showing subgrain boundaries perpendicular to the lineation and to the [001] axis of the crystals. (b) In G3, twisted crystal showing the initiation of recrystallization along the kink planes. (c) In G3, dynamic recrystallization of small new grains in parent crystals showing subgrain boundaries.**

Garnet (0.5-19 vol. %) has a slightly variable composition from one sample to the other (Table S2) and no significant zoning was observed. It is almost exclusively almandine (Alm)-pyrope (Prp) solid-solution (Alm = 0.55-0.62, Prp = 0.3-0.42, $X_{Fe2+}=$ 0.55-0.67) with relatively constant minor contents of Grossular (Grs) (0.05-0.07) and Spessartine (Sps) (0.01-0.03). These compositions fall in the field of Beni Bousera Grt (Alm $_{0.5-0.8}$, Prp $_{0.1-0.3}$, Grs $_{0.03-0.25}$, Sps $_{0-0.06}$, $X_{Fe2+\ 0.6-0.8}$; El Maz and Guiraud, 2001). Some crystals (e.g., in G3) contain inclusions of Rt, ilmenite (Il), graphite (Gph), Plg, KF and Qz. Grt crystals have shapes varying from one sample to the other depending on fracturing and breakdown (e.g., Fig. 3). Garnet breakdown resulted in the formation of reactional coronas whose width is variable depending on samples (Fig. 3 and 5a). For instance, the replacement of garnet is minimal in sample G3 and maximal in G2 in which initial garnet crystals were reduced to tiny residual grains (<1 vol. %) dispersed inside their products of destabilization (G2-Fig. 3). However, Grt initial shape was probably rounded to ovoid (e.g., G3 Fig. 3). Considering both the volume of reaction coronas and their current modal proportions, the initial Grt content was likely between 20 and 25 vol. % in the QFG.

The destabilization coronas around Grt involve Opx, Sp, minor Iron-Titanium Oxides [ITO = Il + Ti-magnetite (TM)] and subordinated plagioclase and quartz (Fig. 5a). Coronas of samples G2 and G3 involve tiny greyish patches of entangled microcrystals of Qz, feldspars and black osumilite. Os is usually in contact with Plg and its composition ($SiO_2$ = 46 % ± 5, $Al_2O_3$ = 23 % ± 2, FeO = 21 % ± 2, MgO = 3.3 % ± 1.3, CaO = 2.8 % ± 0.9, $Na_2O$ = 1.3 % ± 0.3, $K_2O$ = 1.3 % ± 0.1, Table S3) is close to the Ca-rich Os initially described by Miyashiro (1956; $SiO_2$ = 48.3 %, $Al_2O_3$ = 22.1 %, FeO = 13.5 %, MgO = 9.1 %, CaO = 5 %, $Na_2O$ = 0 %,  $K_2O$ = 0 %). The modal proportion of Opx (5-10 vol. %) in garnet coronas, determined using EBSD mapping, is likely underestimated because most crystals forming these reaction coronas (Fig. 5) are too small (≤50µm) and imbricated with smaller Sp grains (10-20 µm) to be correctly indexed at the scale of a thin section. Opx crystals shape is essentially acicular; they are greenish to brown in color with a well-marked pleochroism. Their compositions are significantly variable within each individual sample but, on average, quite similar from one sample to another (Table S4). For instance in



the case of Opx in kelyphite auround garnet, the $FeO_{Total}$ average content is 28.5 % ($\sigma = 0.7$) in G2, 27.8 % ($\sigma = 2.4$) in G3, 27 % ($\sigma = 1.7$) in G4, and the $Al_2O_3$ average content is 11.9 % ($\sigma = 0.5$) in G2, 11.1 % ($\sigma = 3.8$) in G3, 13 % ($\sigma = 2.8$) in G4. Such high Fe and Al contents suggest that Opx result from Grt breakdown under high temperature conditions (Harley, 1998). The $X_{FeTotal}$ ($Fe_T$ / $Fe_T$ + Mg) varies from 0.4 to 0.61. The $Fe_2O_3$ content of Opx in the kelyphitic coronas computed using Schumacher's (1991) method is variable but low; the ratio ($100*Fe^{3+} / Fe^{2+}$) reaches at most 15, except for one Opx in the glass in G3 which reached 27 (Table S4).


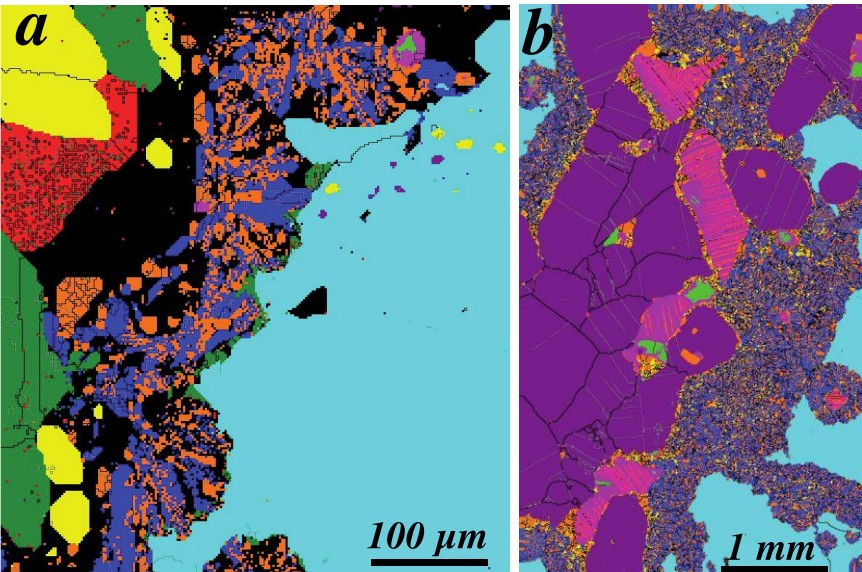

**Figure 5: Detailed EBSD maps showing destabilization coronas around garnet: (a) in the quartzo-feldspathic granulite G3 (~100-150 µm wide), and (b) in the restitic TAF 501 (~1 mm wide). In G3 the garnet displays corrosion embayments and is surrounded by a first rim of tiny plagioclase grains (green) along its boundary then by a second rim corresponding to a mixture of spinel (orange) and orthopyroxene (blue). In TAF501, garnets also display embayments and the coronas around them only involves orthopyroxene, spinel and scarce minute quartz grains (yellow). Sillimanites (purple) are surrounded by a necklace of spinel and quartz, and ilmenite (fuchsia) contains relics of rutile (light green) and exsolution of magnetite (orange). In both maps, domains in black are not indexed.**


Dark-brown to black small spinel grains are present in all quartzo-feldspathic samples (6-9.5 vol. %) and are most often dispersed inside the destabilization coronas around Grt (Fig. 5a). These Sp are hercynite solid solutions ($X_{FeTotal} = 0.57\text{-}0.75$; $Al_2O_3$ = 51-62 wt %; Table S5), always with low contents of $TiO_2 + Cr_2O_3 + MnO + ZnO$ (< 2.7 %). The $X_{Fe}^{2+}$ computed using Droop's (1987) method, lies between 0.77 and 0.95. In addition, in sample G2, Sp grains form discontinuous necklaces around 225 Sil crystals.



Accessory minerals in the Tafraoute QFG consist of rutile (Table S6), ilmenite (Table S7), graphite and zircon, as in Beni Bousera. The most abundant is Ilmenite (0.4-8.6 vol. %). Minute crystals of rutile (0.1-0.6 vol. %) are also dispersed through the rock and some are included in garnets; they contain ~99 % of TiO2, <0.6 % of ($Al_2O_3 + Fe_2O_3$), and 2630 ppm of Zr in G2. Graphite is both dispersed in the rock and included in garnet. Zircon occurs in the matrix and is also present in garnet coronas,
but it is absent inside garnet.

Glass in G2 appears as small patches inside garnet coronas and has a rather homogeneous composition ($SiO_2$ ~63 %; ($Na_2O$ + $K_2O$)~6 %; $Al_2O_3$ ~15 %; $FeO_{Total}$ ~10 %; MgO ~1.3 %; CaO ~0.5 %; Table S8). Glass is less abundant in G3 and G4 and relatively richer in CaO (~5 and 6.5 % respectively) and in $FeO_{Total}$ (~14 and 17 %), and it is poorer in alkalis ($Na_2O$ + $K_2O$ = 2.5 and 0.5 % respectively), in $SiO_2$ (56 and 53 %), while Al and Mg contents are almost similar (Table S8).

In the QFG, the chemical composition of K-feldspars is relatively homogeneous ($0.72 < Xor < 0.82$; $0.16 < Xab < 0.26$, Table S9). In contrast, the composition of plagioclases is variable from one sample to the other (Table S9): for G2 and G3 Xan falls between 0.32 and 0.42. In G2 Xab is comprised between 0.37 and 0.57, and 0.61 in G3. In these two samples, Xor are close ($0.06 < Xor < 0.11$) except for two analyses in G2 (Xor = 0.17 and 0.27). In G4, plagioclases are richer in Na (Xab ~0.8) and poorer in Ca (Xan ~0.2) with Xor values around 0.05.

Feldspars and quartz make it possible to differentiate various microstructures (Figure 3a, b and c). Sample G2 (Qz 25 vol. %, KF 11 vol. %, Plg 2.8 vol. %, Table 1; Fig. 3a) differs in the presence of almost monomineralic aggregates (1-2 mm thick) of K-feldspars parallel to the foliation, involving coarse- and fine-grains. The coarsest grains (up to 5 mm) are highly perthitic lenticular porphyroclasts displaying undulose extinction and surrounded by polyhedral smaller grains (0.2-0.5 mm) free of internal microstructure. These smaller grains display frequent 120°-triple junctions and are generally free of perthitic
exsolution; they have the same composition than the large KF crystals between exsolutions. Tiny grains (~100 μm) of plagioclase crystallized between these smaller KF, either along their boundaries or at triple junctions. In addition, a few larger grains (up to 500 μm) of plagioclase ($Ab_{57}$, $An_{35}$, $Or_8$) are dispersed in KF aggregates. Quartz is rather abundant in this sample; it appears mostly as large and usually elongated xenomorphic crystals (up to 2.5 mm long) forming quartz-rich domains between K-feldspar and sillimanite-rich layers. Some large grains display a weak undulose extinction, but there is no evidence
of annealing.

In sample G3 (Qz 15 vol. %; KF 10 vol. %; Plg 17 vol. %, Table 1; Fig. 3b), in addition to the preferred shape orientation of sillimanite, the foliation is underlined by more or less continuous, almost monomineralic layers (0.5-2 mm thick) of fine- to medium-grained plagioclase (100-500 μm) or quartz (100-700 μm), both showing a tendency towards polygonal shapes. Locally, quartz forms ribbons up to 5 mm long in which quartz grains display a rectangular shape (platten-quartz; Fig. 3b)
suggesting grain boundaries migration. In sample G3, but also in G4, quartz locally fills embayments and cracks in garnet, suggesting late percolation of a Si-rich fluid-phase. KF grains (<250 μm) are locally included within Plg or Qz layers. In addition, discontinuous necklaces of tiny plagioclase grains (~10μm) occur between garnets and their surrounding corona. A peculiarity of G3 is the presence of Qz, KF and Plg as micro-inclusions within some garnets. The composition of these Plg



(Xan ~0.34, Xab ~0.58, Xor ~ 0.09) is faintly poorer in Na compared to the one of Plg outside Grt (Xan ~0.33, Xab ~0.61, Xor
~0.07), while compositions of KF (Table S9) are almost similar (Xor ~0.77; Xab ~0.2; Xan ~0.02).

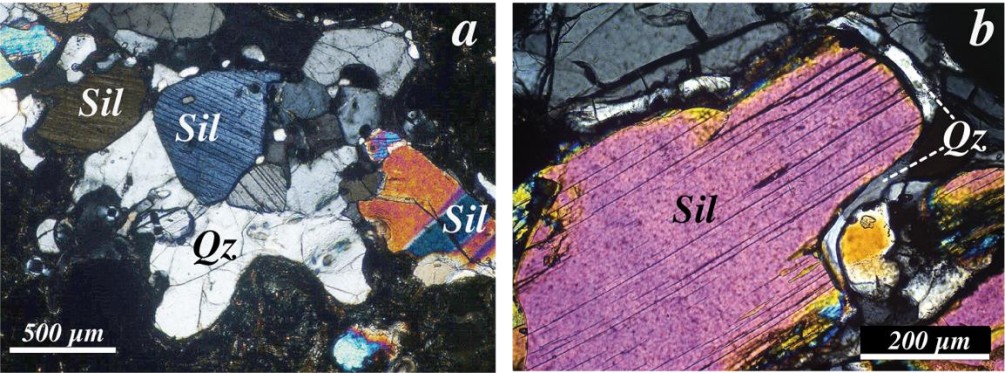

**Figure 6: Photomicrograph in polarized-analyzed light of undeformed interstitial quartz crystal (a) and quartz films**
**around sillimanite (b) in sample G4.**

Sample G4 (Fig. 3; Qz 22 vol. %; KF 9 vol. %; Plg 7 vol. %, Table 1) displays a rather homogeneous microstructure. In
this sample, quartz grains (50-200 µm) are free of intracrystalline substructure; they are either dispersed or concentrated in
layers up to 3 mm wide. In these layers, quartz crystals are frequently elongated with their long axes oblique to the foliation
(Fig. 3). The largest quartz grains have an interstitial habitus (Fig. 6a), while smaller ones display a polygonal shape with
frequent 120° triple-junctions. In addition, narrow films (<10 µm wide) of quartz occur along boundaries of some sillimanite
and feldspars (Fig. 6b). As in G3, most garnets display embayments and fractures filled either by microgranular quartz or Opx
and Sp or by a mixture of these three minerals. Millimeter-sized aggregates of KF and Plg (Xab = ~0.77; Xan = ~0.18 and Xor
= ~0.05) interstitial grains are locally dispersed within Qz aggregates. This network wraps both garnets (≤6 mm) surrounded
by reactional coronas as in G3 and prismatic sillimanite. Altogether, these observations point to a late percolation of a Si-, Na-
, K-rich fluid-phase throughout the rock.

**4.2 The restitic granulites (RG)**

Restitic granulites (Fig. 2b) differ from the QFG by the presence of: 1) abundant sillimanite displaying similar chemical
composition (Table S1) and evidence of intracrystalline deformation than in QFG, 2) a widespread anastomosed network of
Opx and Sp, and 3) low proportions (vol. % < 4) and small sizes of quartz and feldspars, and 4) many large pockets (up to 1
mm) of brown glass locally forming a diffuse network superimposed to the Opx and Sp network. In TAF501, Sil crystals (46
vol. %; Table 1) are contiguous and display an elongated prismatic shape (up to 3 mm long and 1 mm wide). As supported by
crystallographic orientation measurements presented below in section 5 "**Crystal preferred orientations (CPO)**" and Fig. 10



and S6, these Sil crystals display a CPO defining the foliation and lineation. Most Sil crystals are surrounded by discontinuous
girdles of spinel (~50 µm in size) and quartz (10-20 µm) grains, suggesting a destabilization of Sil leading to the crystallization
of these two phases (Fig. 3 and 5b). In TAF500, when Sil crystals form aggregates, Sil-Sil boundaries are rather rectilinear,
locally with 120° triple-junctions and free of evidence of destabilization (Fig. 3 and S4). On the contrary, external boundaries
are polylobate and display embayments. In this sample, continuous coronas of Sp (100-300 µm) and Qz (20-50 µm) grains
systematically wraps Sil isolated crystals and aggregates and this mixture also fills their embayments (Fig. 3 and 5). The largest
embayments are frequently neighboring ilmenite crystals, and Sp+qz coronas also surround these ilmenite crystals, suggesting
that ilmenite is involved in the destabilization of Sil. Locally, small grains of corundum (≤ 350 µm, $Al_2O_3$ = ~97.5 %, FeO =
~2 %, Table S10) are alongside sillimanite boundaries and even fill embayments. This supports that, locally, the Sil
destabilization may have generated Crn through the reaction Sil ⇋ Crn + Qz. According to Guiraud et al. (1996a), Ouzegane
et al. (2003) and Diener and Powell (2010), this reaction occurs at temperature between 800 and 1100 °C under pressure near
1 GPa.

In both studied TAF 500 and 501, garnets are pluri-millimetric to centimetric in size and have similar chemical
compositions than in the QFG samples ($X_{alm}$ = 0.51 and 0.56 respectively, $X_{pyr}$ = 0.41 and 0.36, $X_{sps}$ = 0.02, $X_{grs}$ = 0.05 and
0.06, $X_{Fe2+}$ = 0.56 and 0.61; Table S2). In addition, they display a skeletal shape resulting from intense corrosion. In TAF501,
Grt are surrounded by an anastomosed microgranular network composed of Opx (100-300 µm) and spinel (~50-250 µm)
anhedral grains mixed with tiny quartz grains (≤30 µm). This network has locally infiltrated between sillimanites. In TAF500,
the anastomosed network has the same mineral composition, but differs by the size and shape of Opx that surround residual
garnets. Orthopyroxene (10.2 vol. %) mostly appears as large poikilitic monocrysts (Fig. 7), sometimes >0.5 cm long, with
extremely irregular shapes. This habitus suggests an interstitial crystallization likely from a liquid. The composition of Opx
around garnet in both RG (Table S4) falls in the same variation domain as for QFG. Their averaged $Al_2O_3$ content is 13.7 (σ =
1.9) in TAF500, and 11.5, (σ = 1.5) in TAF501. The averaged FeO content is 24.6 (σ = 0.7) in TAF500 and 27.4 (σ = 0.9) in
TAF501. The averaged $X_{FeT}$ is 0.43 for TF500 and 0.51 for TAF501.



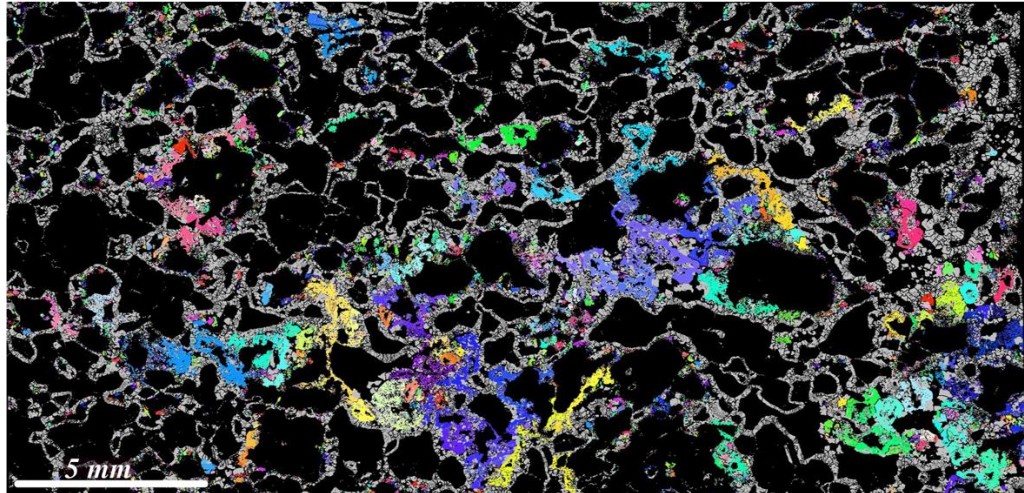

**Figure 7: Detailed EBSD map of restitic granulite TAF500: all constitutive minerals are in black except spinel in grey,**
**while all other colors correspond to orthopyroxene. These colors depend on the crystallographic orientation (Euler angles) of each Opx crystal. This map reveals the habitus of small grains of spinel that form coronas around sillimanite and ilmenite, and the peculiar habitus of Opx. The uniformity of color (and, thus, of crystallographic orientation) over large areas substantiate that most Opx crystals are rather large in size and undeformed. Their sprawling habitus and the abundance of spinel and sillimanite inclusions support an interstitial to poikiloblastic crystallization.**


TAF500 is free of K-feldspar and plagioclase while TAF501 locally contains some grains of plagioclase (0.8 vol. %; 50 - 250 µm; Table 1) with $X_{an}$ varying between 0.22 and 0.39 (Table S9). Opaque minerals proportions vary significantly between the two selected RG (Table 1): in TAF500 only Il (~8.6 vol. %) and in TAF501 Rt + Il + TM (respectively 1, 1.9, 0.4 vol. %). In these samples, opaque minerals are either dispersed between sillimanite and garnet or embedded in the fine-grained mixture
forming the anastomosed network. Zr content in rutile varies between 1900 and 2350 ppm, but there is no significant variation between core and rim of single crystals. In TAF501, iImenite appears as small to medium sized anhedral crystals, scattered throughout the rocks (Fig. 3). The largest crystals (up to 1 mm) frequently contain residual cores of rutile (≤ 250 µm) and exsolution lamellae of Ti-magnetite (Fig. 5b). When ilmenite is in contact with the kelyphitic coronas of garnet, it displays embayments and is bounded by a continuous strip of Ti-magnetite (~10 µm wide). In TAF500 (Fig. 3 and S4); Ilmenite crystals
are larger (up to 2mm) and are free of TM exsolutions as stated above, these crystals are usually surrounded by Sp aggregates and, thus, separated from Sil or Opx.

Brown-glass (± vesicular; Fig. 8) appears as ameboid pockets (estimated vol. %: 5-10) in the two RG (up to 1.5 mm wide in TAF500, and 0.5 mm in TAF501). Compared to glasses of QFG, those found in the RG are relatively richer in FeO, MgO, TiO$_2$ and less rich in SiO$_2$, Na$_2$O and K$_2$O (SiO$_2$ ~48 % in TAF500 and ~52 % in TAF501; Na$_2$O + K$_2$O ~ 0.2 % and ~0.65 %;
Al$_2$O$_3$ ~16.4 % and 16.1; FeO$_{Total}$ ~18.5 % and ~20.2 %; MgO ~3 % and ~2.8 %; CaO ~5.8 % and ~4.8 %; Table S8). Fibro-radiated aggregates (up to 600 µm) of acicular Opx (± minute spinel and quartz), centered either on an opaque grain or a patch



of residual brown-glass, are present in both RG samples (Fig. 8). Such a microstructure evokes spherulitic crystallisation of a glass. In addition, in TAF500, some larger crystals of OPX in contact with glass pockets show overgrowth leading to the formation of idiomorphic terminal faces inside glass.


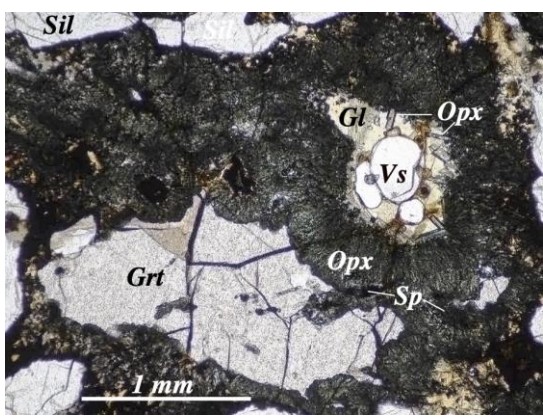

**Figure 8: Microphotograph of a vesicular glass pocket (TAF501) in the center of a fibro-radiated aggregate of orthopyroxene (Opx). Gl = glass, Grt = garnet, Sil = sillimanite, Sp = spinel, Vs = vesicle.**

### 4.3 Corundum megacrysts

A peculiarity of Tafraoute tuffs is the presence of hexagonal tablets of dark blue to black corundum idiomorphic megacrysts (up to 3 cm in diameter and 1 cm thick; Fig. 9a). Such crystals have not been encountered in the studied granulite xenoliths where corundum was only observed in the refractory granulite TAF500 as small (< 200 μ) xenomorphic and interstitial crystals. A few megacrysts have clean flat faces, while most of them display more or less irregular faces due to the presence of corrosion cavities often filled with brown palagonite. In thin section, although

the crystals appear irregularly colored (Fig. 9b), their cores are generally more intensely colored (light blue to slightly purplish) than their rims which additionally appear always darkened (Fig. 9b). Furthermore, they display multidirectional polysynthetic twins (Fig. 9c) and they are deformation free but strongly fractured especially at their peripheries (Fig. 9b). Face corrosion, fracturing and rim darkening are likely consequences of their transport in lava from their place of origin to the surface.

One megacryst, selected for its size (1 cm) and transparency (Fig. 9), has been analyzed for major and trace element (Table S10). Its Al and Fe contents seem slightly higher at the core ($Al_2O_3$ = ~98 % and FeO = 2.1 %) compared to the rims (~97 and 1.7). Note that these Fe contents are slightly higher than that of the interstitial corundums of restitic sample Taf 500 (0.9 < FeO % < 1.1). This megacryst contains anhedral micro-inclusions of titano-magnetite ($TiO2$ = ~20 %, Table S10) and sillimanite grouped in a small millimetric cluster and with

compositions close to those encountered in refractory xenoliths. In addition, a small Opx crystal with very high Al-



contents ($Al_2O_3$ = 22.5 %) is stuck on one of its faces (Table S10). Such very high Al contents are uncommon (to our knowledge, they have been found only once in Opx from Indian Eastern Ghats Province pelitic granulites, Bhattacharya and Kar, 2002), but they are nevertheless close to those measured in some Opx encountered inside the keliphittc corona of G3 garnets or in the glassy thin coating around keliphitized garnets in G2 sample ($Al_2O_3$ up to

360    19.3 %).

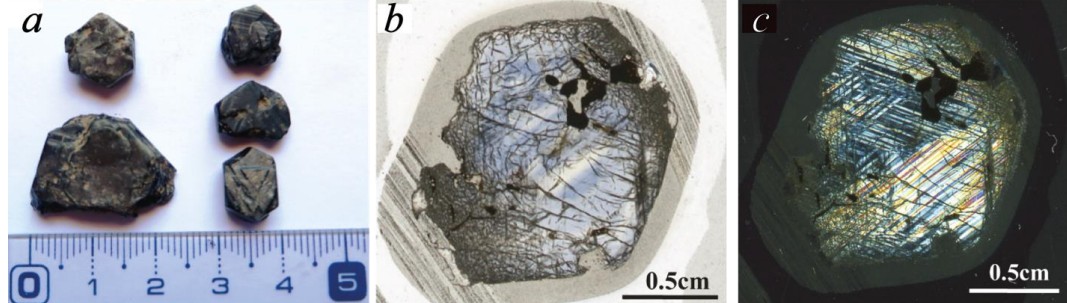

**Figure 9: Corundum megacrysts from Tafraoute tuffs (a). Microphotographs of the megacryst selected for analysis in plane-polarized light (b) and in cross-polarized light (c). On figure (a), the analyzed megacryst is the upper left crystal.**


The origin of the Tafraoute corundum megacrysts is problematic: is it magmatic or metamorphic? The origin of magmatic corundum from basalts has been discussed in several articles (e.g. Peucat et al 2007, Uher et al., 2012 and references therein). All authors consider that, in addition to Fe, Mg, Ti, Cr contents, the Ga content is essential for determining corundum origin. Peucat et al. (2007) even demonstrated that the Ga/Mg ratio, in conjunction with the Fe concentration, is an efficient tool to

discriminate metamorphic from magmatic corundum. To use this discrimination criterion, we carried out 4 ICP-MS–LA analyses on the selected megacryst (Fig. 9). Analytical procedures and analyze results are available in Table S10.

According to Peucat et al. (2007), magmatic blue corundum found in alkali basalts are commonly medium-rich to rich in Fe (with average contents between 2000 and 11000 ppm), high in Ga (> 140 ppm,), and low in Mg (generally > 20 ppm) and high Ga/Mg ratios (> 10). The metamorphic blue sapphires are characterized by low average Fe contents (< 3000 ppm), low

Ga contents (< 75 ppm), and high Mg values (> 60 ppm) with low average Ga/Mg ratios (<10). The Tafraoute megacryst has a high Fe average content of 14770 ppm (σ = 1470) and it is also Ga rich (325 ppm, σ = 19); its Mg content is very variable (53 ppm, σ = 53) and its Ga/Mg ratio is consequently variable but relatively low (6 in average) and therefore lower than those proposed by Peucat et al. (2007) for the magmatic corundum. Its average contents in Ti and Cr are 459 ppm (σ = 159) and 196 ppm (σ = 80) respectively. In the binary Fe vs. Ga/Mg and the triangular (Fe - Mg*100 - Ti*10) diagrams proposed by Peucat

et al., (2007) our megacryst plots in the magmatic field but close to the metamorphic field limit. In the diagrams (Cr*10 – Ga/Mg*100) of Sutherland et al. (2009) modified by Uher et al. (2012), it plots in the metamorphic field. These first results obtained on only one sample suggest a possible mixed origin for the Tafraoute corundum megacysts.



## 5 Crystal preferred orientations (CPO)

The crystallographic preferred orientation of the main constitutive minerals of the five selected granulite samples
have been analyzed using EBSD measurements.

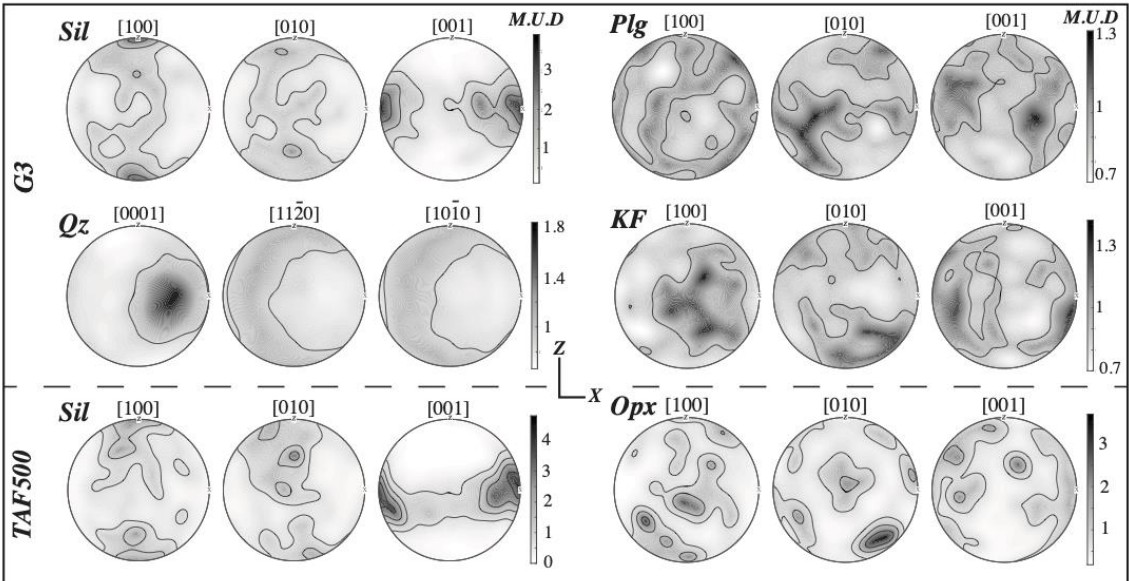

**Figure 10: Representative Crystallographic Preferred Orientation (CPO) of: a) sillimanite, quartz, K-feldspar and plagioclase in the quartzo-feldspathic granulite G3, b) sillimanite and orthopyroxene in the restitic**
**granulite TAF500. Lower hemisphere stereographic projections with contours at 1 Multiple of a Uniform Distribution (MUD) intervals. The grey scale corresponds to MUD for each cell. To facilitate comparisons, CPOs have been rotated to put [001] of sillimanite in the EW direction considering that the maximum concentration of this axis is representative of the lineation. The foliation is ~EW and normal to the plane of the pole figure. In both samples, sillimanite displays a clear preferred orientation despite a weak dispersion of**
**[001] in the foliation plane. In G3 feldspars do not show any reliable preferred orientation and quartz displays a weak one without correlation with the CPO of sillimanite. In TAF500 orthopyroxene is randomly oriented. Additional data are available in supplementary data Fig. S6 and S7.**

### 5.1 Restitic granulites (Fig. 10 and Fig. S6)

In TAF 500 and TAF501, only Sil, Grt and Opx are abundant enough to provide reliable CPO measurements. The
large sillimanite crystals display substructures typical from dislocation-creep and have developed a moderate but well-defined crystallographic fabric. The presence of weakly deformed or even undeformed small grains may weaken the




fabric strength of sillimanite. The CPO is characterized by a concentration of [001] axes around a direction parallel to the foliation and [100] and [010] dispersed in a wide girdle normal to the maximum of [001]. This girdle contains a weak concentration of [100] and [010] normal to the foliation plane. This [001]-fiber CPO type, together with the

presence of subgrain boundaries normal to [001], suggests the dominant activation of the [001] dislocation glide possibly on both the (100) and (010) slip planes (e.g., Piazolo and Jaconell, 2013). However, participation of oriented crystallization or inheritance from pre-existing kyanite crystals to the formation of this preferred orientation cannot be ruled out.

Garnets are almost randomly oriented and no CPO can be defined. They show several point concentrations with

no correlation between them (see Fig. S6), which are likely due to the separation of original crystals in several pieces resulting from fracturing and garnet breakdown.

Opx [100] and [001] only display weak concentrations without any correlation with the sillimanite CPO.

## 5.2 Quartzo-feldspathic granulites (Fig. 10 and Fig. S7)

Sillimanite crystals also display a weak but well-defined CPO similar to the one in the restitic granulites. This fabric

likely results from the ductile deformation at the origin of the intragranular microstructure and dynamic recrystallization of large sillimanite crystals, as previously described.

Quartz, in samples G2 and G3, displays a weak concentration of [0001] in a direction oblique to the main concentration of $[001]_{sil}$ and the $<a>$ and $<m>$ axes tend to form girdles corresponding to a dispersion around a plane normal to the main concentration of [0001]c (Fig. 10). In sample G4, the quartz CPO is more random and not reliably

interpretable (Fig. S7). This weak fabric might be due either to a faint deformation through the activation of the $<a>$-slip direction on prismatic planes, or to oriented crystallization from a percolating melt.

K-feldspars and plagioclases, display nearly random orientation, even when they form almost monomineralic layers. This lack of preferred orientation, consistent with the frequent interstitial habitus of these phases, points to a late crystallization from a percolating melt.

## 6 Geothermobarometry

Considering the mineralogical and microstructural characteristics of the Tafraoute granulites, only garnet core with its inclusions (rutile, ilmenite, graphite, plagioclase and K-feldspar) and the core of KF porphyroclasts in G2 can be reliably considered as belonging to a primary paragenesis. Due to its prismatic habitus and the presence of 74° cleavages in some crystals, sillimanite probably results from later polymorphic transformation of primary kyanite. In

this hypothesis, primary paragenesis of the Tafraoute and Beni Boussera granulites might have been similar. According to El Maz and Guiraud (2001) estimates, the P,T equilibrium conditions of Beni Boussera granulites reached 1-1.3 GPa and 800-870 °C. In this hypothesis the granulites from Tafraoute would have been initially





equilibrated at a depth of ~35-40 km, close to the limit between the kyanite and sillimanite stability field. On the other hand, Opx and Sp coronas observed in the Tafraoute samples are probably the products of a later destabilization of garnet through the reaction: Grt + Sil/Ky ⇆ Al-Opx + Sp ± Plg ± Qz (Fig. 5) and will be considered as representing a second paragenesis. Quartz and feldspars from the matrix display evidence of late remobilization and recrystallization possibly related to partial melting. The glass, and thus the spherulitic Opx around glass pockets, likely result from flash melting of rocks triggered by their entrapment and ascent in lavas followed by fast cooling at the surface.

**6.1 Equilibrium conditions for the Primary Paragenesis P I (Table 2)**

The lack of biotite in the studied samples excludes application of garnet-biotite to estimate P, T conditions corresponding to the first paragenesis. It is also not possible to use the feldspars pairs in the matrix because they likely were remobilized during their subsequent evolution. However, the garnets in sample G3 include plagioclase, K-feldspar, rutile, quartz, ilmenite and graphite, which can be used to estimate the temperature and pressure corresponding to the P,T conditions of the primary paragenesis.

**Table 2. Summary of temperature (°C) and pressure (GPa) estimates for the primary paragenesis of G3 granulite, using garnet core and feldspars included in garnet. Temperature is calculated at 1.1 GPa, and pressure at 870 °C. To calculate P,T conditions, we used the software of Wen and Nekvasil (1994, model of Nekvasil and Burnham, 1987), Putirka (2008), Yavuz and Yavuz (2022, model of Benisek et al., 2010, for ternary feldspars: albite, orthoclase and anorthite) and the calibration of Price (1985). Pressures calculated at assuming temperature of 870°C, using the calibration of Caddick and Thompson (2008) and Essene's abacus (1989) for GASP and GRIPS barometers. The mixing models of Koziol and Newton (1989) and Newton and Haselton (1981) were used for garnet and plagioclase respectively. In all samples, the stable aluminosilicate considered is kyanite.**



| Primary paragenesis: Feldspars in garnet (G3) | | | |
|---|---|---|---|
| Temperature (°C at 1.1 GPa) | | | |
| Plg - KF | 17 - 14 | 18 - 21 | 20 - 10 |
| Price (1985) | 863 | 853 | 876 |
| Wen and Nekvasil (1994) | 895 | 846 | 859 |
| Putirka (2008) | 864 | 860 | 862 |
| Yavuz and Yavuz (2022) | | | |
| T albite | 864 | 871 | 890 |
| T orthose | 854 | 891 | 896 |
| T anorthite | 889 | 867 | 845 |
| Pressure (GPa at 870°C) | | | |
| Plg - Grt | 17 - 36 | 18 - 48 | 20- 17 |
| GRIPS (Essene 2008) | 1.05 | 1.07 | 1.07 |
| GASP (Essene 2008) | 1.18 | 1.2 | 1.22 |
| Caddick and Thompson (2008) | 1.00 | 1.03 | 1.04 |

### 6.1.1 Pressure

The mineralogical assemblage: garnet-plagioclase-aluminosilicate-rutile-ilmenite-quartz, is well known for its
geobarometric potentiality. Several barometers and calibrations have been proposed for this assemblage but they are
all sensitive to temperature. Considering the lack of biotite in the primary paragenesis of our samples, likely resulting
from the total consumption of this mineral through fluid-absent melting (at temperatures ranging between 850-875
°C, Vielzeuf and Holloway, 1988), and the similarity with the Beni Bousera granulites (Fig. 11), a starting reference
temperature of 850 °C can be used to estimate the pressure. In these conditions, the pressures obtained using the
465 GRIPS (garnet-rutile-ilmenite-plagioclase-silica) and GASP (garnet-aluminosilicate-silica-plagioclase) abacus of
Essene (1989) with garnet inclusions and garnet core of sample G3 are between 1.1 and 1.2 GPa (Table 2). The
calibration of Caddick and Thompson (2008) for GASP barometer provides pressures slightly lower: ~1 GPa at 850
°C (Table 2). These estimates are in good agreement with Bohlen et al. (1983) experimental results carried out on the
GRIPS assemblage (1.2 and 1.3 GPa at 850°C and 900 °C).





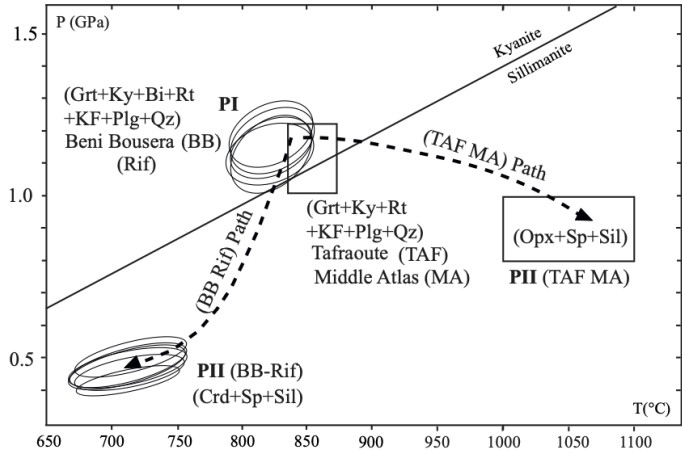

**Figure 11: P-T-t diagram showing the primary (P I) and secondary (P II) parageneses of Tafraoute granulites from Middle Atlas (TAF MA) and Beni Bousera (BB-Rif). In the case of Rif granulites, the pressures and temperatures of the primary and secondary paragenesis have been determined using THERMOCALC program of Holland and Powell (1988), the ellipses correspond to 2σ error.**

### 6.1.2    Temperature

The binary feldspar thermometers (Price, 1985; Putirka, 2008), weakly sensitive to pressure, and ternary feldspar using SolvCalc 2.0 software (Wen and Nekvasil, 1994) can be applied to the two feldspars included in garnet. With the reference pressure of 1.1 GPa estimated above, these thermometers give temperatures in the range 850-880°C (Table 2 and Fig. 11). These temperature estimates are similar to those obtained using Yavuz and Yavuz (2022) software.

These P, T estimates are in good agreement with conventional petrogenetic grids (e.g., Vielzeuf and Holloway, 1988; Nair and Chacko, 2002 and references therein), the lack of biotite and the presence of rutile in the primary mineralogical assemblage. These results corroborate that these rocks were equilibrated within the stability field of kyanite, near the kyanite-sillimanite phase boundary (Fig. 11).

### 6.2 Equilibrium conditions of the secondary paragenesis P II (Table 3)

Estimates of the temperature and pressure were obtained using the Opx-Grt geothermobarometer, considering the Opx from the coronas in contact with the rims of garnets. This geothermobarometer is based on $Fe^{2+}$-$Mg^{2+}$ exchange equilibrium, therefore, the $Fe_2O_3$ content in the phases involved in these exchange reactions should be estimated. Hence, as mentioned above, the Schumacher's (1991) method was used for the Opx.





**Table 3: Summary of temperature and pressure estimates for the secondary and tertiary parageneses (PII and PIII). Temperature (°C) was calculated at assuming pressure of 0.9 GPa and pressure at assuming temperature of 1100 °C. PII: using Opx in contact with garnet rim. Garnet-orthopyroxene geothermobarometer: Harley (1984), Aranovich and Berman (1997), Sudholz et al. (2022), Harley and Green (1982) and Nickel and Green (1985). PIII: using minerals in glass, calibrations and programs used: Orthopyroxene-Ilmenite (Charlier et al., 2007); Orthopyroxene-Spinel (Sato et al., 2012); Orthopyroxene-Glass (Beattie 1993, and equation 28b of Putirka, 2008); Silica activity in Glass (Albarède, 1992); Fe-Ti-oxides and $Log_{10}$ ($fO_2$) (Lepage's ILMAT program, 2003: Spencer and Lindsley (1981) calibration and Lindsley and Spencer model (1982) for the ulvospinel and ilmenite molar fractions in the oxides). $Log_{10}$ ($fO_2$) = $Log_{10}$ of oxygen fugacity.**

| Paragenesis II | | | | | | | | | | | | |
|---|---|---|---|---|---|---|---|---|---|---|---|---|
| **Samples** | **G2** | | **G3** | | **G4** | | **TAF 500** | | | **TAF 501** | | |
| **Garnet - Orthopyroxene** | 15-10 | 64-13 | 9-54 | 8-8 | 70-66 | 62-56 | 1-7 | 4-15 | 64-6 | 9-38 | 36-23 | 77-24 |
| | | | | | *Temperature in °C (at 0.9 GPa)* | | | | | | | |
| *Harley 1984* | 1114 | 1126 | 1093 | 1000 | 1153 | 1069 | 1091 | 1137 | 1079 | 1048 | 1136 | 1090 |
| *Aranovich and Berman (1997)* | 1117 | 1108 | 1089 | 1068 | 1123 | 1086 | 1086 | 1098 | 1064 | 1112 | 1092 | 1069 |
| *Sudholz et al. (2022)* | 1063 | 1071 | 1030 | 939 | 1107 | 1013 | 1080 | 1084 | 1020 | 1065 | 1089 | 1043 |
| | | | | | *Pressure in GPa (at 1100°C)* | | | | | | | |
| *Harley and Green (1982)* | 0.76 | 0.78 | 0.75 | 0.82 | 0.74 | 0.77 | 0.77 | 0.77 | 0.77 | 0.82 | 0.80 | 0.76 |
| *Nickel and Green (1985)* | 1.04 | 1.06 | 1.20 | 1.05 | 0.91 | 0.91 | 0.96 | 0.96 | 0.98 | 0.93 | 1.05 | 1.05 |
| **Paragenesis III** | | | | | | | | | | | | |
| **Samples** | **G2** | | **G3** | | **G4** | | **TAF 500** | | | **TAF 501** | | |
| | | | | | *Temperature in °C (at 0.9 GPa)* | | | | | | | |
| **Orthopyroxene - Ilmenite** | 5-28 | | 13-26 | | | | 9-8 | 70-71 | | 2-42 | 76-41 | |
| *Charlier et al. (2007)* | 1032 | | 1029 | | | | 1094 | 1098 | | 1142 | 1080 | |
| **Orthopyroxene -Spinel** | 16-34 | 32-71 | | | 18-63 | 67-61 | 31-45 | 68-72 | | 18-14 | 76-15 | |
| *Sato et al. (2012)* | 1070 | 1108 | | | 1091 | 1057 | 1131 | 1119 | | 1147 | 1065 | |
| **Orthopyroxene -Glass** | 5-35 | 9-58 | 60-26 | | 9-15 | 10-7 | 9'-8 | 9-10' | 70-69 | 2-38 | 3-39 | 38-21 |
| *Beattie (1993)* | 1152 | 1086 | 1124 | | 1110 | 1115 | 1105 | 1120 | 1154 | 1157 | 1154 | 1172 |
| *Equation 28b (Putirka 2008)* | 1011 | 1012 | 1038 | | 1055 | 1049 | 1077 | 1114 | 1160 | 1117 | 1118 | 1122 |
| **Glass (silica activity)** | 35 | 58 | 26 | 30 | 7 | 15 | 8 | 10' | 69 | 21 | 38 | 39 |
| *Allbarède (1992)* | 1004 | 1011 | 1020 | 1047 | 1027 | 1030 | 1044 | 1071 | 1109 | 1074 | 1071 | 1072 |
| | | | *Oxides : T°C and $Log_{10}$ (fO2)* | | | | | | | | | |
| **Samples** | **G2** | | | | **G4** | | | | | **TAF501** | | |
| **Ilmenite - Titanomagnetite** | 22-30 | | | | 59-55 | | | | | 9-8 | | |
| *Spencer and Lindsley (1981) Lindsley and spencer model (1982)* | 1160 | -7.99 | | | 1081 | -8.66 | | | | 1181 | -8.46 | |

### 6.2.1 Temperature

This thermometer is almost insensitive to pressure. Several calibrations (Harley, 1984; Aranovich and Berman, 1997 and Sudholz et al., 2022) have been used. With a reference pressure ~1 GPa, temperature estimates computed for all QFG and RG samples fall between 1020 and 1120 °C (Table 3).

These values point to an increase in temperature after the primary paragenesis (Fig. 11). This temperature increase, corresponding to the breakdown of Grt, is consistent with the polymorphic transformation of kyanite into sillimanite.



### 6.2.2 Pressure

The Grt-Opx geobarometer used in this study is temperature sensitive and therefore an error of 50 °C generates a variation of 0.14 GPa. Using the calibration of Harley and Green (1982) with reference temperatures of 1050 and 1100 °C, the estimated pressures are 0.76 and 0.88 GPa respectively (Table 3). With the Nickel and Green (1985) calibration, at the same temperatures, relatively higher pressures were obtained (0.86 – 1.25 GPa respectively; Table 3). The assemblage osumilite, orthopyroxene, garnet, feldspars, sillimanite and quartz (as in sample G3), is stable at

T >850 °C between 0.8 and 0.9 GPa (Harley, 2008 and references therein) and 0.7 and 0.85 GPa (Das et al., 2001). Therefore, for Tafraoute granulites, the pressure estimates obtained using Nickel and Green (1985) calibration are likely slightly overestimated. In this case the temperature increase suggested by our estimates would be coeval with a slight pressure decrease (from ~1.1 to ~0.9 GPa).

### 6.3 Temperature estimates using matrix K-feldspars and plagioclase (Table 4)

Quartz and feldspars from the matrix display evidence of late remobilization and crystallization possibly related to a partial melting event. The temperatures computed using Plg and KF of the matrix would therefore correspond to equilibrium conditions subsequent to those recorded by PII. Plg-KF temperatures were estimated from the exchange of Ca, Na and K using Price (1985) geothermometer and Wen and Neckvasil (1994), Putirka (2008) and Yavuz and Yavuz (2022) softwares. For the three QFG analyzed in this study, temperature estimates between 720 and 830°C

have been obtained (Table 4). These values are significantly lower than those obtained for both primary and secondary parageneses; they might result from the cooling of the lower crust after the episode of heating recorded by the second paragenesis.

**Table 4: Summary of temperature estimates (°C) calculated at 0.9 GPa using matrix K-feldspars**
**and plagioclases for G2, G3 and G4 samples. Calibrations and softwares used are the same as in table 2.**



| | T°C at 0,9 Gpa | | | |
|---|---|---|---|---|
| | G2 | | G3 | G4 |
| | Plg39-KF64 | Plg70-KF65 | Plg44-KF45 | Plg46-KF60 |
| | A | B | B | B |
| Price (1985) | 1012 | 855 | 797 | 723 |
| Wen & Nekvasil (1994) | 1030 | 876 | 833 | 789 |
| Putirka (2008) | 834 | 854 | 827 | 789 |
| Yavuz and Yavuz (2022) | | | | |
| $T_{albite}$ | 1200 | 866 | 788 | 628 |
| $T_{orthose}$ | 1087 | 882 | 821 | 611 |
| $T_{anorthite}$ | 744 | 832 | 851 | 923 |

## 6.4 Temperature estimate for the RG glass formation and crystallization (Table 3)

The glass present in the studied xenoliths represent the last thermal event underwent by these granulites. To estimate the temperature of this late event, we analyzed the glass and the minerals resulting from its partial crystallization (Opx, Sp and ITO). The geothermometers used are: Opx-Il (Charlier et al., 2007), Opx-Sp (Sato et al., 2008), Opx-glass (Putirka, 2008), silica activity in glass (Albarède, 1992) and Iron-Titanium oxides (ILMAT software; Lepage, 2003). This last geothermometer is independent of the pressure; the others are almost insensitive to the pressure since an error of 0.4 GPa causes an error of less than 35 °C. Most of the calculated temperatures at a reference pressure of 0.9 GPa, fall in the range between 1020 and 1160 °C (Table 3). The corresponding calculated Log $f_{O2}$ are tightly between -8 and -8.67 (Table 3) supporting that the glass formation occurred under oxidizing conditions. These estimated conditions may therefore result from the incorporation of granulitic xenoliths in the host basaltic magma.

## 7 Discussion

Two types of Al-rich granulite have been defined among the xenoliths from the quaternary Tafraoute maar: "refractory" (almost free of quartz and feldspars) and "quartzo-feldspathic" (up to 42 vol. % quartz + feldspars). Both types contain prismatic sillimanite deformed through dislocation creep, garnet surrounded by undeformed destabilization coronas of variable width. The coronas around garnets involve Opx, Sp and minor Qz and Plg tiny crystals. This likely reflects the reaction Grt + Sil ⇆ Al-Opx + Sp, classically attributed to temperature increase (higher than 940 °C in the KFMASHTO system, Wheller and Powell, 2014). In addition, the quartzo-feldspathic granulites contain quartz, plagioclase and K-feldspar often concentrated in almost monomineralic layers.

In sample G2, K-feldspar layers contain eye-shaped perthitic porphyroclasts displaying evidence of intracrystalline deformation, surrounded by smaller, K-feldspar polyhedral grains. These small grains are most often exsolution-free, and likely result from dynamic recrystallization of porphyroclasts. This supports that these porphyroclasts: 1) belong to the initial paragenesis, 2) underwent high-temperature deformation resulting in partial





recrystallization of their rim. Samples G3 and G4 contain layers of plagioclases with polygonal grain shape; this supports that they were annealed and might also, at least partially, derive from primary crystals. However, as stated before, some K-feldspar and plagioclase grains, free of intragranular deformation, locally display interstitial habitus supporting a post-deformation crystallization from a melt.

In all QFG samples, quartz also displays contrasting shapes and habitus that likely represent: 1) original large grains (e.g., G2), 2) annealed, possibly dynamically recrystallized smaller crystals, sometimes forming ribbons (e.g., G3), 3) deformation-free interstitial quartz with irregular grain shapes (e.g., G2 and G4), which locally forms thin films between preexisting phases, or infills garnet embayments and fractures (e.g., G4 and G3).

This variety of habitus and microstructure of feldspars and quartz likely results from several successive tectono-metamorphic events. These three phases were present in the initial paragenesis, then, during the subsequent heating event, they underwent crystal-plastic deformation and partial melting at the origin of local percolation and interstitial crystallization. In such a scenario, the refractory granulites might represent domains where the quartzo-feldspathic components were almost totally melted and extracted.

The evolution of the Tafraoute lower crust proposed below is substantiated by both mineralogical observations and P-T conditions estimated from successive parageneses recognized in the studied granulite samples (Fig. 11):

- The observation of cleavages typical of kyanite in some prismatic sillimanites, suggests that the granulites of Tafraoute were originally equilibrated in the kyanite stability field such as those of Beni Bousera (~850 °C and ~1.2 GPa). This interpretation is consistent with the P,T equilibrium conditions (1.1 ± 0.1 GPa and

850-880 °C) estimated from garnet and its inclusions (Rt + Il + Gph + KF + Plg + Qz). This suggests an initial depth of ~40 km for these granulites. Such conditions are compatible with fluid-absent melting, in agreement with the lack of biotite in the studied QFG samples. Considering the similarities with Beni Bousera granulites, these conditions likely correspond to the Hercynian orogeny (Fig. 11).

- P,T estimated from garnet and the phases resulting from its destabilization are ~0.9 ± 0.1 GPa and 1050-

1100 °C, suggesting that the lower crust underwent a temperature increase of ~200 °C compared to the primary equilibrium temperature. This heating could have been coeval with a slight pressure decrease (~0.2 GPa corresponding to ~7 km of crustal thinning). This temperature increase, probably progressive, may have been responsible, in a first time, for the transformation of kyanite into prismatic sillimanite. Then, this transformation was followed by the moderate deformation of newly crystallized sillimanites through

dislocation creep and dynamic recrystallization. This heating may also account for the crystallization of Ca-rich osumilite in the G2 and G3 granulites, likely resulting from the reaction Grt + Feldspar + Qz $\leftrightarrows$ Os + Opx as proposed by Audibert et al. (1995). In our case, osumilite is Ca-rich (up to 4 wt %) and the feldspar implicated in the reaction is probably plagioclase. Approaching peak temperature, anhydrous partial melting may have been triggered. This melting episode may be responsible for melt percolation followed by





crystallization of interstitial quartz and feldspars in the QFG matrix, and possibly also for the formation of restitic granulites.

-   Equilibrium temperatures estimated from feldspars in the QFG matrix, are significantly lower (around 750-830 °C) than those recorded by the two previous parageneses. These low temperatures probably correspond to the cooling of the lower crust after the peak temperature of the second stage.

The restitic granulites likely underwent a similar sequence of events as the QFG, but with different intensities. Assuming that these rocks were initially quartzo-feldspatic, the first part of their evolution was likely similar to the QFG granulites, especially regarding the transformation of kyanite into sillimanite at the onset of the heating episode. The restitic mineral composition of these rocks together with their structure support that they have undergone a subsequent episode of partial melting leading to the almost complete consumption and extraction of quartz and

feldspars. The resulting lack of quartz and feldspars in the RG may have favored the reaction between sillimanite and garnet. The peculiar microstructure of TAF500 (lobate shape of sillimanites and continuous spinel corona around them, skeletal shape of garnet, interstitial Opx wrapping garnets) might be explained by an intensification of the reaction between garnet and sillimanite after the consumption of quartz and feldspars.

     Two hypotheses can be proposed to explain the formation of the RG. In the first one, the extraction of the quartzo-

605 feldspathic component would have occurred due to the partial melting triggered in the QFG granulites by temperature increase (~1100 °C) at the end of the ultra-high thermal event 2. The second hypothesis considers that the thermal event responsible for the formation of RG results from the Middle Atlas plio-quaternary volcanic activity. In the late case, the RG might represent restitic walls of basaltic dykes emplaced before the entrapment of xenoliths by the eruption of Tafraoute maar, as proposed in the cartoon of Fig. 12. The formation of Sp coronas around sillimanite and

610 of the poikilitic large interstitial crystal of Opx present in TAF500 might be associated to this event, perhaps after the melting of the quartzo-feldspathic phases and extraction of the resulting melt. The percolation of an $H_2O$- and $CO_2$-rich and SiO2-undersaturated liquid issued from the dykes might have triggered, in a first time, an almost complete *in situ* dissolution of the Opx from the kelyphitic corona around garnet (Shaw, 1999), associated with a peripheral dissolution of sillimanite (and possibly also of ilmenite rims in contact with the melt). Shortly thereafter, this

dissolution may have been followed by crystallization, also in situ, of poikilitic Opx and formation of spinel coronas around sillimanite.



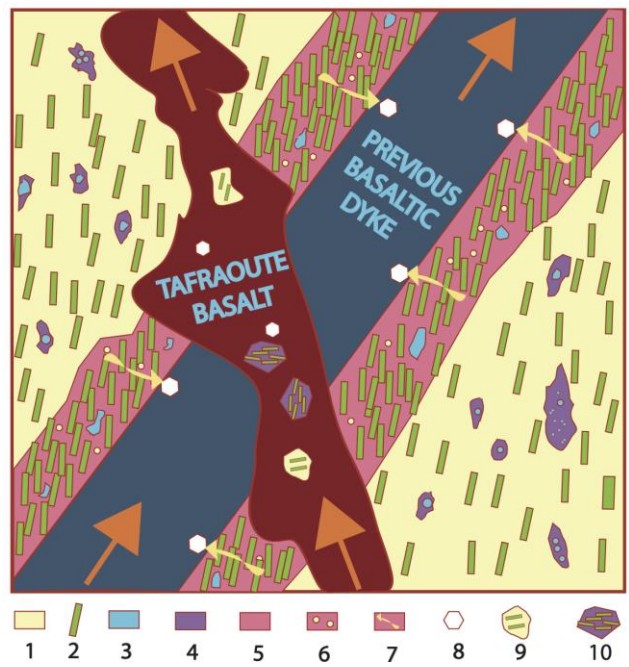

**Figure 12: Cartoon (not to scale) showing the possible relationships in depth between the different rocks and minerals found as xenolith and megacrysts in the Tafraoute tuffs (1) quartzo-feldspathic granulite, (2) prismatic sillimanite, (3) garnet, (4) kelyphitization corona, (5) restitic granulite, (6) anhedral corundum in restitic rock; (7) melt issued from partial melting of the dyke wall, (8) corundum megacryst, (9) quartzo-feldspathic granulite xenolith, (10) restitic granulite xenolith.**

The presence in the RG, and in a lesser extend in the QFG, of pockets of vesicular, more or less dark brown glass containing minute crystals (Al-Opx, Sp, Il and TM), points to a final episode of flash melting followed by fast cooling. This late flash melting is probably linked to the entrapment and fast transport of xenoliths up to the surface by Tafraoute basanite (650 Ka). The equilibrium temperature between 1020 and 1150 °C estimated from the minerals inside the glass agrees with this hypothesis. This temperature is akin to those usually measured for basaltic magma.

## 7.1 The corundum megacrysts

How to integrate the crystallization of idiomorphic corundum megacrysts in the evolution of the Tafraoute granulites remains an open question. As previously indicated, the chemical composition of the studied megacryst combines both magmatic and metamorphic characteristics. Their crystallization was probably initiated inside restitic granulites, as supported by the presence of small and anhedral corundum crystals in some restitic samples and the presence of several metamorphic phases (including sillimanite) as inclusions inside corundum megacrysts. Since corundum was





not encountered in the studied quartzo-feldspathic granulites, this mineral likely formed at the same time as the granulites acquired their refractory characters through overheating and felsic phases melting. But the acquisition of large size and automorphy by megacrysts probably requires further free growth in liquid. As proposed above, the restitic granulites might represent walls of alkali basaltic dykes emplaced before the Tafraoute eruption (Fig. 12). The

percolation of $SiO_2$-undersaturated and $H_2O$-$CO_2$ rich fluid issued from lava dykes inside the restitic walls might have led to inter-element exchanges and desilication reactions favoring corundum formation within the restites. Subsequently, the restitic walls were disintegrated by ascending magma and the corundum crystals released from their rock matrix. The Si and Al rich melt resulting from the melting of walls felsic phases might have been simultaneously incorporated into the lava. The incorporation of such a melt into a Si-undersaturated alkali magma would favor the

corundum growth and acquisition of idiomorphy, a process already suggested by Coenraads et al. (1990) and Giuliani et al. (2009). But, in our case, this scenario is only based on the study of a single megacryst. A study on a larger number of samples will have to be carried out to have a more precise idea of the origin of the Tafraoute corundum megacrysts.

## 7.2 Microstructures and textures

Both types of granulites display a foliation essentially marked by the orientation of sillimanite and, in the QFG, by quartz and feldspars layers. The large prismatic crystals of sillimanite display uncommon evidence of intracrystalline deformation: subgrain boundaries normal to [001], fan-like crystals and recrystallized new grains (Fig. 4). This substantiates a moderate crystal-plastic deformation of these high-strength prismatic sillimanite after their crystallization. The microstructure formed during this episode was subsequently modified by the UHT partial melting.

The crystallographic preferred orientation of sillimanite, although rather weak, is well-defined and displays a concentration of [001] parallel to the foliation, which marks the lineation (Fig. 10). Together with subgrain boundaries dominantly orthogonal to [001], this preferred orientation supports that [001] was the dominant slip direction of dislocations during deformation.

Garnet crystals display scarce evidence of solid-state deformation: weak bending of the crystal lattice,
elongated shape, and fractures. In both granulite types, garnets are partially to almost totally destabilized (Fig. 3, 5) and the products of its breakdown do not display evidence of solid-state deformation nor any reliable CPO (Fig. 10 and Fig. S6-S7). This strongly supports that these rocks were not deformed after the heating responsible for garnet breakdown, and, thus, that the latter occurred after the deformation of sillimanite.

Quartz in the felsic layers does not display evidence of internal solid-state deformation but it sometimes
displays evidence of annealing and grain boundary migration (straight boundaries and 120° triple junctions). Similarly, small grains of K-feldspar frequently show evidence of annealing and grain boundary migration. However, in sample G2 (Fig. 3), perthitic K-feldspars porphyroclasts displaying a faint undulose extinction are surrounded by



small KF grains following their irregular boundary. These aggregates of small KF grains also involve tiny plagioclase grains. This microstructure is suggestive of dynamic recrystallization leading to crystallization of albite-rich Plg from
perthites. In addition, in several samples such as G4, most quartz and KF crystals have an interstitial habitus and are locally clustered in elongated aggregates oblique to the foliation. These observations are pointing to a crystallization from a percolating melt after deformation has stopped. Plagioclase displays similar microstructures than quartz and K-feldspars, although there is less evidence of late crystallization from the percolating melt from which quartz and K-feldspar interstitial grains crystallized.

In sample G2 and G3, quartz displays a weak [0001]-axial CPO poorly correlated with the sillimanite CPO. In both cases, the maximum concentration of [0001] is in the foliation plane but is variably oblique (30-45°) relative to the sillimanite [001] axis. Two processes may account for this weak CPO: 1) preservation of some quartz crystals inherited from the initial paragenesis; they have undergone the same solid-state deformation than sillimanites and escaped partial melting, and 2) oriented crystallization from a melt under a moderate stress oriented differently than
the one responsible for the deformation of sillimanite. Sample G4 does not display any reliable preferred orientation of quartz, in agreement with the interstitial habitus of this phase, and, thus, with crystallization from a percolating melt under static conditions.

In all studied samples, KF and Plg do not display any reliable preferred orientation. This is in good agreement with a post-kinematic crystallization of most feldspars from a melt. Nevertheless, the presence of inherited
porphyroclasts in some samples supports that this melt would result from late partial melting of primary quartzo-feldspathic layers.

Altogether these observations support that the evolution of the Tafraoute granulites occurred during a progressive temperature increase (Fig. 11) followed by cooling. In the first time, the temperature increase coeval with strain were responsible for the transformation of kyanite in prismatic sillimanite, as suggested above, and for its
deformation. This deformation likely ended before the high temperature breakdown of garnet as supported by the lack of deformation of destabilization products. In this scheme, the crystallization of interstitial quartz and feldspars grains observed in the QFG likely resulted from the subsequent crystallization of a percolating melt due to partial melting of original quartzo-feldspathic layers. Indeed, considering the peak-temperatures recorded by these samples, cooling of the lower crust was probably slow and, thus, crystallization of melts was delayed.

The microstructural and mineralogical peculiarities of the restitic granulites suggest that they reacted more intensely to a thermal event (either associated to the paragenesis 2, or to the emplacement of basaltic dikes in the lower crust). Despite this evolution, internal deformation and CPO of sillimanites have been preserved.

**7.3 Comparison of the Tafraoute and Beni Bousera lower crust evolution (Fig. 11)**



The primary paragenesis recorded by the Tafraoute QFG was probably similar to the Beni Bousera Hercynian granulites one (El Maz and Guiraud, 2001), suggesting that it is also Hercynian in age. The initial likeness of the Beni Bousera and Tafraoute lower crust is strengthened by the similarity of the shallower lithospheric mantle from the Rif range to the Middle Atlas before the Alpine orogenic event (Pezzali et al., 2015). However, the Tafraoute QFG differ from Beni Bousera ones by the presence of deformed large prismatic sillimanite, Opx + Os in Grt coronas, the lack of biotite and secondary cordierite and by evidence of subsequent heating to ultra-high temperature. These differences support that the Hercynian QFG from Tafraoute and Beni Bousera, initially almost similar, have undergone drastically different subsequent evolutions (Fig. 11). The post-hercynian evolution of Beni Bousera granulites is characterized by an almost adiabatic decompression of ~0.8 GPa due to their exhumation during the alpine orogeny (El Maz and Guiraud, 2001; Álvarez-Valero et al. 2014; El Bakili et al. 2020; Rossetti et al. 2020, Fig. 11). Contrastingly, the Tafraoute granulites were not exhumed and recorded a temperature increase (~200 °C) of the lower crust, possibly associated with a slight P decrease (≤0.3 GPa).

Altogether, these results suggest that the lower crust under the North Morocco, at least between the Rif and the Middle Atlas, underwent rather homogeneous metamorphic conditions during the Hercynian orogeny, as already suggested by Rossetti et al. (2020). This favors the hypothesis that the temperature increase (~200 °C) suffered by the Tafraoute lower crust was related to a post-hercynian event likely pre-alpine or alpine.

**7.3 A possible scenario for the evolution of the Tafraoute lower crust**

At the location of the future Middle Atlas belt, the Alpine deformation was preceded by a Triassic-Jurassic episode of rifting linked to the Central Atlantic opening (e.g., Michard et al., 2008 and references therein). In the tabular Middle Atlas, where is located the Tafraoute maar, the alpine deformation was limited to local faults reactivation (Gomez et al., 1996; Zeyen et al. 2005; Frizon de Lamotte et al. 2008; Saura et al. 2014). This favors the hypothesis that the post-hercynian tectono-metamorphic evolution of the Tafraoute lower crust was rather related to the Mesozoic pre-alpine rifting.

In this hypothesis, the Tafraoute Hercynian crust (around 40 km thick and with temperatures in the lower crust likely 850-880 °C at the end of the Hercynian orogeny) may have been slightly thinned by ~7 km, simultaneously with an increase in temperature of ~200 °C compared to the temperature recorded by the granulites garnet inclusions. This high temperature increase for such a limited thinning, may be due to two possibly complementary processes: 1) thinning of the crust associated with mantle upwelling, and 2) emplacement of gabbro bodies in, or just below, the lower crust. This second process is supported by the presence of many undeformed subalkali gabbro xenoliths in the Tafraoute tuffs, which are alike to Triassic to Cretaceous gabbros intrusions reported in many places within the Atlas belt (e.g., Hailwood and Mitchell, 1971; Smith and Pozzobon, 1979; Westphal et al., 1979; Calvin et al. 2017; and





references therein). This scenario is consistent with the mineralogical, microstructural and CPO data and the continuous evolution presented above.

**8 Conclusion**

The tuffs of the quaternary Tafraoute maar (Tabular Middle Atlas, North Morocco) contain two types of lower crustal sillimanite- and garnet-rich metapelitic granulites: layered quartzo-feldspathic and restitic ones. Combining the new

data obtained from these granulites with those acquired from mantle xenoliths sampled in the same maar and with available geophysical data on the Middle Atlas, we propose an evolution of the lower crust beneath the Middle Atlas. This evolution would involve two tectono-metamorphic events followed by an ultimate flash-heating and fast decompression related to the maar eruption:

-   During the first event, the lower crust acquired its foliation and yield equilibrium P, T conditions of 1.1 ±

0.1 GPa and 850-880 °C similar to the ones of lower crustal granulites from the Rif. Under these conditions, the Al-sillicate present in the rocks was likely kyanite and the primary paragenesis also comprised garnet, rutile, plagioclase, perthitic K-Feldspar, quartz, graphite and ilmenite. As supported by the absence of biotite, these rocks probably underwent fluid-absent partial melting. This event may correspond to the Hercynian orogeny as this was established in the Rif belt.

-   The second tectono-metamorphic event is characterized by a progressive reheating up to ultrahigh temperatures (1050 ± 50 °C) under slightly lower pressure conditions (0.9 ± 0.1 GPa). This would have first triggered the transformation of kyanite into sillimanite. After its crystallization, sillimanite experienced a moderate crystal-plastic deformation that likely affected also quartz and feldspars. This deformation was followed by stress relaxation under persisting ultra-high temperature metamorphic conditions that initiated

garnet-breakdown through the reaction Grt + Sil ⇋ Al-Opx + Sp (± Plg ± Qz). Reaching peak temperatures, anhydrous partial melting of felsic layers occurred in quartzo-feldspatic granulites and the resulting melt spread through the rocks. Then, the temperature progressively dropped to ~800°C, leading to static crystallization of deformation-free, interstitial quartz and feldspars from the melt. This second tectono-metamorphic event probably results from Triassic-Jurassic rifting, possibly coeval with underplating of

gabbroic magma. In this hypothesis, considering the lack of deformation subsequent to the crystallization of quartz and feldspars from the melt produced by the second tectono-metamorphic event, the lower crust of the tabular Middle Atlas was not significantly affected by the alpine, Cretaceous-Eocene, compression. This hypothesis agrees with the preservation of geochemical characteristics typical of extension in the upper mantle (El Messbahi et al., 2015).

-   Two hypotheses may explain the formation of the restitic granulites: 1) they might be due to the UHT melting event through almost total consumption of quartz and feldspars and extraction of the resulting melt, and 2)



they might represent the walls of basaltic dykes emplaced during the long-lasting (Miocene to quaternary) volcanic activity in the Middle Atlas. The studied RG might represent pieces of these walls extracted as xenoliths.

-    The last event, recorded by all studied granulites, results from their entrapment in the host basanitic lava and their very fast transfer to the surface by the Tafraoute maar eruption 650 ka ago. During this event, the sampled granulites have been heated up to 1150 °C, producing vesicular glass that partially crystallized during the fast decompression and cooling.

## Authors contributions:

AEM and JMD sampled the studied granulites. In addition, AEM performed observations under optical microscopes, made chemical analysis, computed Temperature and Pressure estimates, AV organized EBSD data acquisition, performed data processing and computed EBSD maps and crystallographic preferred orientations. JMD performed additional chemical analysis and comparisons of mineralogical determinations with EBSD data. All authors contributed equally to the writing of the manuscript and the preparation of figures and tables.

**Competing interests**: The authors have no conflict of interest with other researchers or laboratories.

## Acknowledgements:

This study was performed as part of a collaborative multidisciplinary research project on lower crust rocks from Morocco involving the Faculty of Sciences of Meknes (University Moulay Ismail, Morocco) and Geosciences Montpellier (CNRS & University of Montpellier, France). This research has been partially funded by the International Lithosphere Program CC4-MEDYNA. We thank Jean Louis Bodinier for financial support through MEDYNA, Houssa Ouali, Fleurice Parat, Andréa Tommasi and Hicham El Messbahi for their help in the field or for their analytical expertise.  We thank Olivia Mauguin and Fabrice Barou for their assistance during Microprobe and EBSD analyses respectively and Christophe Nevado and Doriane Delmas for the preparation of high-quality thin sections.
.

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
