# Peer review of "Post-hercynian ultra-high temperature tectono-metamorphic evolution of the Middle Atlas lower crust (Central Morocco) revealed by metapelitic granulites xenoliths"

_EGUsphere, 2024_

## Referee Comment (RC2)

**FSBM**
FACULTÉ DES SCIENCES BEN M'SIK
UNIVERSITÉ HASSAN II DE CASABLANCA

**Faouziya Haissen**
**full professor, Petrology and Geochemistry**
**Department of Geology**

Casablanca, 06th June 2024

To: Editor of Copernicus Publications

**Review of the paper**: Post-hercynian ultra-high temperature tectono-metamorphic evolution of the Middle Atlas lower crust (Central Morocco) revealed by metapelitic granulites xenoliths

**Author(s):** Abdelkader El Maz et al.

**Manuscript number**: egusphere-2024-508

**Dear Editor,**

The paper is well written with good data, especially the part where authors use the petrographic an geothermobarometric data to compare the evolution of granulites in two different tectonic contexts: the Rif and the Middle Atlas belt which can constrain the evolution of the middle crust in North Africa.

The language is properly and correctly used except few paragraphs which can be reorganized and reduced. All the important comments are listed below:

1- The figure 1 (and in my opinion all figures) could be highly improved, the authors talk about the North Middle Atlas fault which is not indicated on the figure, its location can be deduced but readers not familiar with the geology of Morocco can be lost;

2- There are paragraphs and descriptions which require significant improvement in the English style as for example the use of "Several tens" on line 107…;

3- Petrographic photos are lacking in the manuscript, the authors use time to time few photos to indicate some textures, but I believe that studying such nice rocks require the use of good photos showing the beautiful reactional textures of these rocks;

✉ Faculté des Science Ben M'sik Avenue Cdt Driss El Harti, BP 7955 Sidi Othmane Casablanca, Maroc
☎ 212(0)522704671/72/73; 🖷 212(0)522704675

4- At the beginning of the paper, the authors describe the presence of coronas around garnet without explaining the reaction that led to this reactional texture. The authors explain briefly such reaction at the end of the paper;

5- The Cartoon of the figure 12 is not clear for me;

6- I am really confused about the composition of garnet, authors describe it as homogeneous compositions without zoning even in the garnet is highly resorbed and replaced by the coronitic minerals. It is the case of all garnet analyzed and how many profiles did you done on garnets?

7- The section about geothermobarotry of the studied rocks must be improved and the authors must explain to the reader what compositions of garnet is used to calculate the first and the second events of the evolution adopted for these rocks, which plagioclase was used (inclusion, matrix or the reactional texture one…). If the garnet is homogenenous in composition, there is no problem but if it is zoned, the composition used in calculation (rim or core) can affect strongly the P-T calculated..

Reviewing all those comments will improve highly the quality of the manuscript .

Professor Haissen Faouziya
Professor at Hassan 2 University of Casablanca,
Coordinator of undergraduate studies of
Geology at the faculty of Sciences Ben M'sik
faouziya.haissen@gmail.com
+212661253824

---

## Author Response (AR2)

*Reply to Reviewer #1 comments*

We first want to warmly thank reviewer #1 whose comments and suggestions have been very helpful. We have considered all suggestions and try to provide answers to the questions. We hope the changes made on the manuscript are satisfactory.

1)    Petrography: textures and microstructures are not well documented. They are initially presented and discussed by means of EBSD maps, but a complete set of photomicrographs for the two rock types is needed.

*We have added two supplementary figures showing photographs of two representative thin sections, one of the G3 quartzo-feldspatic granulite and the other of the TAF501 refractory granulite.*

[Figure]

G3                                    TAF501

2)    Bulk rock analyses (major and some trace elements) would help constrain better the processes responsible for the formation of these two rock types and the interaction between walls of dykes and fluids

*We agree that this would have improve our data set. Such analyses have not been carried out because there are no adequate facilities in Morocco and the low financing of our research impeded to carry out the analyses in a European Lab.*

3)    P-T estimates are based on conventional thermobarometry which, however, may suffer from important limitations when applied to granulites, in particular for temperature estimates (e.g. Pattison et al., 2003). The authors report analyses for the Zr content of rutile. Why did they not apply the Zr-in-rutile thermometer (Kohn, 2020)? What about calculations of P-T pseudosections considering the melt-reintegration approach?

*Indeed, conventional geothermobarometry has limits when applied to granulites (e.g. Pattison et al. 2003, Harley 1989…), especially for the garnet-Opx couple, and oxides. The calculated temperatures are sometimes underestimated more than 100 °C compared to the real temperature of the paragenesis considered, because of the feedback effect, at the time of cooling of the rock, and often these temperatures correspond to the closing temperatures of the exchange between minerals involved in reaction. Harley (1998), Kelsey and Hand (2015)… indicate that the $Al_2O_3$ contents of Opx (8-12%) is a good indicator of THT (>900 °C) and the Zr contents in rutile constitute a robust geothermometer for the calculation of temperatures for granulites-facies.*

*Temperature and pressure computations in the Tafraoute granulites were performed using armored feldspars in garnet, which compositions has recorded the peak P,T conditions of primary paragenesis. The results are also compatible with fluid-absent melting conditions. Moreover, Kohn (2020) equation using Zr in rutile, gives temperatures close to those obtained using the feldspars included in garnet. For parageneses 2 and 3, the results obtained are in good agreement with experimental data and the high contents of $Al_2O_3$ in the Opx ( ~10-16%), as explained in the text (we did have any underestimating of temperature). This is probably due to the fact that the Tafraoute xenoliths were brought quickly to the surface by very hot basalts and, consequently, cannot develop feedback effect.*

*The construction of pseudosections requires chemical analyzes of the rocks. Unfortunately, as noted above, such analyses have not been carried out. In any case, given the evolution of Tafraoute granulites suggested in our paper, the initial composition of the rocks was probably modified by addition or loss of their quartzo-feldspathic component. This will undoubtedly have more or less significant impact on the position of the univariant reactions and the extents of the stability fields (divariants and trivariants) of the parageneses crossed by the path PT of the rocks.*

Additional comments:

- 51: Bohlen 1991 is not present in the reference list
  o *There was a mistake in the references list. The year of this publication was not 1983 but 1991. This was corrected.*

- 57-58: I suggest rewording for clarity to "UHT metamorphism is marked by the presence of high $Al_2O_3$…."
  o *Done*

- 58: add values for Zr contents in rutile which are considered indicative of UHT metamorphism
  o *In the studied granulites, rutile belongs the first paragenesis (P1) and its content in Zr is significantly lower (1888-2630 ppm) than the Zr content in UHT rutile (>3500 ppm) according to Harley (2008).*

- 61-63: references are needed
  o *Vielzeuf and Holloway, 1988; Vielzeuf et al. 1990 have been inserted.*

- 107: tens are already "several". Delete several
  o *Done*
- 138 etc…: avoid the use of labels as first word…Grt should be Garnet…. Check across the entire manuscript
  o *Correction made*

- 140: rewording: in some QFG and RG samples

- o  *Corrected: " the first one was observed in both QFG and RG samples…"*
- 163: replace "on the opposite" with "on the other hand"
  - o  *Done*
- 240. Not clear. Please reword this sentence
  - o  *Sentence modified: " Various microstructures can be defined from quartz and feldspars*

- 271-272: May these films represent melt pseudomorphs as described by Sawyer (2008) and Holness and Sawyer (2008)?
  - o  *Yes, they result from the percolation of melt along grain boundaries. We have modified the last sentence of the section: "Altogether, these observations support a late percolation of a Si-, Na-, K-rich fluid-phase throughout the rock, and its crystallization in pores and along grain boundaries, evoking the microstructures described by Holness and Sawyer (2008)."*

- 300: not clear, please reword this sentence
  - o  *Sentence modified: " This mixture locally occurs between sillimanite grains."*

- 303: it is not clear why "this habitus suggest an interstitial crystallization likely from a liquid". Please explain better your point here
  - o  *We have modified this sentence: " These monocrysts display extremely irregular shapes, some of them sprawling between sillimanite, ilmenite and/or garnet crystals (Fig. 7). Such a habitus suggests an interstitial crystallization, possibly from a liquid."*

- 331: fibroradiated? I've never seen such a term. Please check in the literature
  - o  *Fibroradiated is used in many articles of petrology and mineralogy. For instance here comes the definition given by the site https://www.le-comptoir-geologique.com/fibroradiated-glossary.html : "This term refers to crystalline textures showing fibers radiating from a center. "*

- 355-360: the authors report very high Al2O3 contents in Opx (>20 wt%). Due to the proximity of corundum, it is reasonable to consider a possible effect of contamination.
  - o  *This section has been modified to answer this question: "In addition, a small Opx crystal with very high Al-contents ($Al_2O_3$ = 22.5 %) is stuck on one face of the corundum megacryst (Table S10). Such very high Al content is uncommon (to our knowledge, they have been found only once in Opx from Indian Eastern Ghats Province pelitic granulites, Bhattacharya and Kar, 2002). In our case, this anomalously high Al content might result from contamination of Opx by the contiguous corundum. However, this Al content is close to those measured in some Opx encountered inside the keliphittc corona of G3 garnets (18.6%) or in the glassy thin coating around keliphitized garnets in G2 sample ($Al_2O_3$ up to 19.3 %), two samples that are corundum free.*

- 382: a possible mixed origin. Please detail better what you mean
  - o  *Sentence modified: " These first results obtained on only one sample suggest a possible origin both metamorphic and magmatic, for the Tafraoute corundum megacryst."*

- 422: reword K-feldsparS and plagioclases as "K-feldpsar and plagioclase crystals"
  - o  *Modification done*
- 438: flash melting. Do you mean incipient melting?

o        *For clarity we have modified the sentence with: " from instantaneous melting (flash melting) of rocks triggered by their entrapment and ascent in lavas"*

• 446: add (PI) after primary paragenesis

o        *Done*

- 508: why is kyanite not preserved into garnet interior? In Ronda granulites (which share a similar story with Beni Bussera granulites) garnet contains Ky in the core and sillimanite toward the rim (see Barich et al. 2014).

*- This remark is fully justified but, unfortunately, we cannot propose a reliable answer. In Beni Bousera granulites, about 5% of the garnets contain a few small kyanite grains in their rim. These garnets are not involved in the reactions forming cordierite and spinel (secondary paragenesis). Those surrounded by a cordierite aureole never contain kyanite.*

*In the case of Tafraoute, if there was some kyanite grains of parageneses 1 included in the garnet rims, due to the post-hercynian temperature increase, there were likely involved in the reaction at the origin of the formation of Opx + Spinel coronas around garnet (as explained in the text). In addition to spinel (an aluminous mineral), aluminous Opx (~10-15% $Al_2O_3$) also crystallized and their high Al-content may result to the fact that the high temperature favors the solubility of Al in Opx (Harley 1998). Garnet in Tafraoute granulites was severely destabilized by the formation of the Opx+spinel coronas and the garnets crystals observable today are residual (see EBSD maps and pictures in our manuscript and also the two microphotographs added to the supplementary figures), so the garnet rims with their possible inclusions of kyanite were probably replaced by the Opx+spinel coronas.*

l. 535-544: The authors apply thermobarometers considering minerals crystallized from the melt now preserved as glass. However, these products are reported as "spherulitic aggregates". How were these tiny spherulitic aggregates analyzed by EMPA? Is it reasonable to consider some degrees of contamination during EMP analysis?

*• Effectively, the phases analyzed inside spherulites are tiny, but for EMP analysis we have selected crystals larger than the diameter of the analytical spot. In addition, we have considered as usable for thermobarometry calculations only analyses for which the sum of major elements oxide % was closed to 100 (i.e., Table S4 Opx, 23 analyses, 98.1<∑<101). If there was contamination by glass, the sum of oxides would be significantly more variable.*

We would like to warmly thank Professor Faouziya Haissen (reviewer #2) for her remarks and comments which greatly helped us to improve the quality of our manuscript. We have taken into account all her suggestions and tried to provide answers to all her questions. We hope that the changes made to the manuscript meet her expectations.

1- The figure 1 {and in my opinion all figures) could be highly improved, the authors talk about the North Middle Atlas fault which is not indicated on the figure, its location can be deduced but readers not familiar with the geology of Morocco can be lost;

- *We have modified Fig. 1 and the new version shows the main faults of Moroccan orogenic belts.*

2- There are paragraphs and descriptions which require significant improvement in the English style as for example the use of **"Several tens"** on line 107;

- *"Several" has been deleted.*

3- Petrographic photos are lacking in the manuscript, the authors use time to time few photos to indicate some textures, but I believe that studying such nice rocks require the use of good photos showing the beautiful reactional textures of these rocks;

- *Two Pictures have been added in supplementary material. They show the two types of rocks in thin sections, both under polarized and polarized/analyzed light. We however favored EBSD maps in the main text since they illustrate very well the mineral composition, microstructure and reactional textures observed in the studied samples (e.g., Figure 5).*
- *In addition, we have added a supplementary figure (S2) showing a garnet crystal initially surrounded by sillimanite crystals, which was almost totally transformed in a kelyphitic corona. This figure shows two microphotographs, one under polarized light and the second under*

*polarized/analyzed light, and the EBSD map of the same area in sample G2.*

4- At the beginning of the paper, the authors describe the presence of coronas around garnet without explaining the reaction that led to this reactional texture. The authors explain briefly such reaction at the end of paper.

*Effectively the reaction that likely led to the formation of kelyphitic coronas around garnet is not presented in the section 4.1 because this section is devoted to the identification of mineral phases present in the studied samples and to the description of their textural relationships. The possible reaction at the origin of the kelyphitic coronas is presented in details at the beginning of the section 7 "Discussion".*

5- The Cartoon of the figure 12 is not clear for me;

- *A short additional comment has been added to the caption of Figure 12 to clarify what we aimed to show: " The plio-quaternary volcanic episode started with the injection of early basaltic dikes that may have been responsible for the partial melting of their walls and formation of refractory granulites (5). Through the reaction between the melt resulting from partial melting (7) of the walls and the basalt of the dykes, corundum megacrysts (8) may have crystallized. Later on, basaltic eruptions such as those of Tafraoute have carried samples of these various materials up to the surface.".*

6- I am really confused about the composition of garnet, authors describe it as homogeneous compositions without zoning even in the garnet is highly resorbed and replaced by the coronitic minerals. It is the case of all garnet analyzed and how many profiles did you done on garnets?

*In the frame of this study, due to our low analytical budget, it was not possible to perform detailed chemical profiles across garnets. Only core and rim analysis have been systematically performed in all studied samples, and no significant chemical variation has been characterized, even in the most resorbed garnets.*

*Another reason which did not encourage us to carry out analysis profiles in garnet is:*

*In the case of metapelites, at low and medium temperatures (i.e., in the presence of muscovite), the garnets generally show a strong growth zonation (the bell-shaped Mn-Ca profil). Compared to the cores, the rims are richer in almandine and pyrope and poorer in grossular and spessartine (Tracy, 1982: Compositional Zoning and inclusions in metamorphic minerals, Reviews in mineralogy, V10, p 355-397).*

*At high temperature (in the presence of K-Feldspar and absence of muscovite, our case), the garnets undergo homogenization by diffusion imposed upon pre-existing garnets or new growth garnets at higher temperature. This homogenization proceeds over an approximate temperature range of 50 °C, with a mean temperature around 640 °C (Yardley, 1977 : an empirical study of diffusion in garnet, American Mineralogist, V62, 793-800), zoning commonly restricted to the outer few tens of micron if it is in contact with a Fe-Mg mineral (Opx, biotite, cordierite...). As a result, and to prioritize the analyzes of other minerals, we limited ourselves to a core analysis and a rim analysis (used in thermobarometry). In two garnets of samples G3 and TAF500, we also did an analysis between rim and core, we found that these garnets are almost unzoned (table below).*

| | G3 (QFG) | | | TAF500 (Restit) | | |
|---|---|---|---|---|---|---|
| | 2 (core) | 6 (towords core) | 3 Rim | 9 (core) | 8 (towards core) | 10 (Rim) |
| Almandine | 0,516 | 0,515 | 0,517 | 0,618 | 0,614 | 0,617 |
| Pyrope | 0,413 | 0,413 | 0,412 | 0,319 | 0,323 | 0,31 |
| Spessartine | 0,018 | 0,017 | 0,017 | 0,014 | 0,015 | 0,015 |
| grossular | 0,053 | 0,054 | 0,054 | 0,048 | 0,048 | 0,058 |

7- The section about geothermobarometry of the studied rocks must be improved and the authors must explain to the reader what compositions of garnet is used to calculate the first and the second events of the evolution adopted for these rocks, which plagioclase was used (inclusion, matrix or the reactional texture one.). If the garnet is homogenenous in composition, there is no problem but if it is zoned, the composition used in calculation (rim or core) can affect strongly the P-T calculated.

*As explained above, garnets are almost unzoned.*

*To estimate the pressure and temperature of primary paragenesis (garnet, kyanite, rutile, K-feldspars, plagioclase, quartz and graphite), we used the garnet core, and the armored feldspars in the same garnet. These feldspars (K-feldspar and plagioclase) are almost unzoned. As noted in the text, only the garnet of the G3 granulite includes feldspars accessible to microprobe analyses, and which were used to estimate primary (Hercynian) paragenesis. For secondary (post-Hercynian) paragenesis, we used the garnet rim in contact with the Opx. For the last event, paragenesis III, we used only the phases included in the glass (Opx in rod and oxides). The analyzes of the phases used in thermobarometry are numbered and indicated in Table 3 in front of each couple (example for G2 sample: Garnet 15 and Opx 10, second determination Garnet 64 and Opx 13). All analyses are available in supplementary tables.*